# A state space modeling approach to real-time phase estimation

Anirudh Wodeyar[1]*, Mark Schatza[2], Alik S Widge[2], Uri T Eden[1,3], Mark A Kramer[1,3]

[1]Mathematics and Statistics, Boston University, Boston, United States; [2]Department of Psychiatry, University of Minnesota, Minneapolis, United States; [3]Center for Systems Neuroscience, Boston University, Boston, United States

**Abstract** Brain rhythms have been proposed to facilitate brain function, with an especially important role attributed to the phase of low-frequency rhythms. Understanding the role of phase in neural function requires interventions that perturb neural activity at a target phase, necessitating estimation of phase in real-time. Current methods for real-time phase estimation rely on bandpass filtering, which assumes narrowband signals and couples the signal and noise in the phase estimate, adding noise to the phase and impairing detections of relationships between phase and behavior. To address this, we propose a state space phase estimator for real-time tracking of phase. By tracking the analytic signal as a latent state, this framework avoids the requirement of bandpass filtering, separately models the signal and the noise, accounts for rhythmic confounds, and provides credible intervals for the phase estimate. We demonstrate in simulations that the state space phase estimator outperforms current state-of-the-art real-time methods in the contexts of common confounds such as broadband rhythms, phase resets, and co-occurring rhythms. Finally, we show applications of this approach to in vivo data. The method is available as a ready-to-use plug-in for the Open Ephys acquisition system, making it widely available for use in experiments.

*For correspondence:
wodeyar@bu.edu

**Competing interest:** The authors declare that no competing interests exist.

## Introduction

Rhythms, consistently periodic voltage fluctuations, are an ubiquitous phenomenon observed in brain electrophysiology across scales and species. Many studies have described a relationship between rhythms and behavior (*Buzsáki, 2006*; *Haegens and Zion Golumbic, 2018*; *VanRullen, 2016*). A prominent feature of rhythms proposed as relevant for neural processing is the phase. Phase has been proposed to coordinate neural spiking locally and potentially, through coherent networks, even globally (*Fries, 2015*; *Maris et al., 2016*). For example, the phase of slow rhythms (3–15 Hz) has shown relationships to dynamics in perception (*Busch et al., 2009*; *Gaillard et al., 2020*; *Gregoriou et al., 2009*; *Helfrich et al., 2018*). Phase-amplitude synchrony is common (*Canolty and Knight, 2010*; *Hyafil et al., 2015*), and has been suggested to change in neurodegenerative disease (*Hemptinne et al., 2013*). Cross-regional, low-frequency phase synchrony is consistently observed as a correlate of top-down executive control (*Widge et al., 2019*). Most work examining the importance of phase for neural dynamics and behavior has been correlative in nature. To better understand the functional relevance of the phase of a rhythm requires an ability to monitor and perturb phase relationships in real-time.

Several methods exist to estimate phase in real-time (*Blackwood et al., 2018*; *Chen et al., 2013*; *Mansouri et al., 2017*; *Rodriguez Rivero and Ditterich, 2021*; *Rutishauser et al., 2013*; *Siegle and Wilson, 2014*; *Zrenner et al., 2020*). At their core, these methods rely on bandpass filtering, a mechanism to forward predict data (using autoregressive [AR] models or linear extrapolation of phase) and the Hilbert transform to estimate the phase. Brain rhythms are often non-sinusoidal (*Cole and Voytek, 2017*) and broadband (*Buzsaki, 2004*; *Roopun et al., 2008*), making development of an accurate bandpass filter difficult. While the contemporary modeling approach using filters accurately

estimates the phase across a range of contexts, several limitations exist that limit the phase estimate accuracy (*Matsuda and Komaki, 2017*; *Siegle and Wilson, 2014*): (1) By depending on bandpass filters, existing real-time phase estimators are susceptible to non-sinusoidal distortions of the waveform, and inappropriate filter choices may miss the center frequency or total bandwidth of the rhythm. (2) Phase resets, moments when the phase slips because of stimulus presentation or spontaneous dynamics, cannot be tracked using filters, which control the maximum possible instantaneous frequency. (3) By filtering the observed data directly, these approaches do not model signal and noise processes separately. Phase estimates thus represent signal and noise phase together. (4) Many existing approaches depend on buffered processing (e.g., Fourier or related transforms), meaning that current phase estimates are delayed in ways that may not support real-time intervention. Further, buffer-based processing is susceptible to edge effects that are not of concern in traditional, offline analysis. (5) Contemporary approaches do not define a measure of confidence in the phase estimate. Despite these limitations, real-time phase simulations using current techniques have produced interesting results (*Cagnan et al., 2017*; *Desideri et al., 2019*; *Hyman et al., 2003*; *Kundu et al., 2014*; *Schaworonkow et al., 2018*; *Siegle and Wilson, 2014*; *Zrenner et al., 2018*), motivating the value of generating better methods to estimate phase causally.

To address the limitations of existing approaches, we implement here a dynamical systems approach to real-time phase estimation. Rather than filtering the data to identify the rhythm of interest, we focus on tracking the analytic signal (a complex valued entity, representing the amplitude and phase of a rhythm *Kramer and Eden, 2016*) as a latent process in the observed time series using a model proposed by *Matsuda and Komaki, 2017*. In doing so, we remove the requirement of bandpass filtering and instead model the signal as a state space harmonic oscillator (or combination of such oscillators) driven by noise. This modeling approach allows: (1) a separate estimate of observation noise, (2) a procedure to model and reduce the confounding activity of other rhythms, and (3) a principled method for assessing uncertainty in the real-time phase estimates. After first fitting parameters of the state space harmonic oscillator model acausally, the method then tracks the phase in real-time assuming spectral stationarity. We call this approach the state space phase estimate (SSPE).

To benchmark the SSPE technique against the current state-of-the-art methods of real-time phase estimation, we design several simulations and analyze case studies of in vivo data. We limit our comparison to the real-time methods proposed in *Zrenner et al., 2020*, and *Blackwood et al., 2018*, methods which provide an estimate of phase on receipt of each new sample of data and have readily available reference code. We additionally apply a standard method for non-real-time phase estimation (an acausal finite impulse response [FIR] filter followed by a Hilbert transform; *Lepage et al., 2013*). We note that these and other existing real-time phase estimation techniques (*Chen et al., 2013*; *Mansouri et al., 2017*; *Rodriguez Rivero and Ditterich, 2021*; *Rutishauser et al., 2013*; *Siegle and Wilson, 2014*) apply a bandpass filter and forecasting technique, and, by that measure, the acausal FIR approach by utilizing more information suggests the room for improvement available for these estimators. We test the ability of these different methods to track the phase of rhythms with different spectral profiles – including narrowband rhythms, broadband rhythms, multiple rhythms, and rhythms with phase resets, and in different signal-to-noise settings. We show that the SSPE method outperforms the existing real-time phase estimation methods in almost all cases. Finally, we illustrate the application of the SSPE method to in vivo local field potential (LFP) and electroencephalogram (EEG) recordings, demonstrate the utility of SSPE for predicting behavior, and introduce a plug-in of the SSPE method for Open Ephys (*Siegle et al., 2017*), making the method widely available for use in real-time experiments. Our work demonstrates that the SSPE method improves the ability to track phase accurately, in real-time, across a diverse set of contexts encountered in data.

## Materials and methods
### State space model framework

To estimate phase in real-time, we utilize a data-driven model (*Matsuda and Komaki, 2017*) that operates based on principles from dynamical systems and Markov models. This type of model (called a state space model) separates what is observed (called the observation equation) from what we wish to estimate (called the state equation). The state equation attempts to capture the underlying, unobserved dynamics of the system, while the observation equation transforms the state into the observed

signal. We generate an optimal prediction for the state using a Kalman filter that compares the initial prediction of the observation with the actual observed value.

Each state variable represents an oscillator with a fixed frequency that undergoes small random perturbations. As these perturbations accumulate, the instantaneous phase and frequency of the oscillator become more uncertain, since the oscillator is not directly observed. We assume that the observed data are noisy versions of these oscillators. The state space model framework combines information from two models: (1) a state model that describes the oscillator dynamics and (2) an observation model that describes how the oscillator generates the data. The methods we develop use both of these sources of information to optimally estimate the instantaneous frequency and phase of each oscillator at every moment in time.

We implement a linear state equation that models the *real* and *imaginary* part of the complex valued analytic signal of rhythms present in the neural data. For each rhythm modeled, the state equation dynamics rotate the analytic signal at that rhythm's frequency. The observation equation sums the *real* part of the analytic signals of every modeled rhythm to predict the observed neural activity. We treat as real-valued the real and imaginary parts of the complex valued analytic signal for ease of analysis and interpretation. This state space model structure allows us to derive the amplitude and phase of each modeled rhythm, and estimate the analytic signal in the temporal domain, thus avoiding complications associated with windowing (i.e., the need to choose large enough window sizes to support sufficient frequency resolution and the associated unavoidable temporal delay in phase estimates) that occur in related frequency domain approaches (*Kim et al., 2018*).

The state space model is linear in the state evolution (dynamics of state transitions) and observation equations. Further, the covariance for the state (which determines the bandwidth for the rhythms) and noise for the observation is Gaussian (white noise). These assumptions (linearity and Gaussianity) imply that we can use the Kalman filter to correct the state estimate based on the true observation (i.e., to filter the state) (*Kalman, 1960*). This type of modeling approach has been successfully applied in many contexts to track latent structure from observed data while accounting for state and observation noise effects (*Brockwell et al., 2004*; *Eden et al., 2018*; *Galka et al., 2004*; *Yousefi et al., 2019*).

To illustrate this approach, we first consider the case of tracking a single, 6 Hz rhythm. The state equation for this scenario is:

$$x_t = aO(\omega_j)x_{t-1} + u_t, u_t \sim N(0, Q)$$

where $x_t$ has both a real and imaginary part. At time instant $t-1$, the modeled rhythm is in state $x_{t-1}$, that is, the rhythm is at some point along its trajectory. The state equation provides a prediction for the next point along this trajectory ($x_t$) by rotating the current state ($x_{t-1}$) and adding noise ($u_t$). The rotation matrix $O(\omega)$ rotates the real and imaginary parts of the state by the amount $\omega$ appropriate for a 6 Hz rhythm:

$$O(\omega) = \begin{pmatrix} \cos(\omega) & -\sin(\omega) \\ \sin(\omega) & \cos(\omega) \end{pmatrix}$$

$$\omega = 2\pi(6/Fs)$$

where *Fs* is the sampling frequency in Hz. After rotation, the state is scaled (or rather, damped) by a factor $a$, and then Gaussian noise (with mean and variance $Q$) is added. We estimate the observation at time t ($y_t$) as the real part of the state at time $t$ with additive noise:

$$y_t = Real(x_t) + \nu$$

where $\nu$ is Gaussian noise with mean 0 and variance $\sigma^2$. Note that this model for the state is akin to a damped harmonic oscillator driven by noise, a model that has been shown to be relevant and useful for gamma rhythms in the visual cortex (*Burns et al., 2010*; *Spyropoulos et al., 2019*) and also for EEG (*Franaszczuk and Blinowska, 1985*).

To model $N$ rhythms, we define $\boldsymbol{X}$ as the state variable, a $2N \times T$ matrix over total time $T$. An individual time point with state $X_t$ ($2N \times 1$ vector) is rotated according to each rhythm's ($x_j^t$, a $2 \times 1$ complex vector for rhythm $j$) central frequency ($\omega_j = 2\pi f_j t \in (0, \pi)$, where $f_j$ is frequency in Hz and $t$ is sampling interval) and damped by a constant ($a_j$). For each rhythm $x_j^t$ we define $\theta_j^t$ as a scalar quantity tracking the phase, and $\boldsymbol{\Theta}$ as an $N \times T$ matrix of phase values for all rhythms over all time. For the

$N$ rhythms, **O** is the rotation matrix, a $2N \times 2N$ and block diagonal matrix, defined by the central frequency of each rhythm. **A** is the scaling matrix, a diagonal $2N \times 2N$ matrix, with scaling factor $a_j \in (0, 1)$ to ensure stability, the same for each rhythm, and different across rhythms. The driving noise for the state $U$ is Gaussian, a $2N \times T$ matrix with covariance structure **Q**, a $2N \times 2N$ diagonal matrix with the same variance for the real and imaginary parts of each rhythm. The state is collapsed into the observation $Y$ (a $T \times 1$ vector) through $M$ (a $1 \times 2N$ vector) which summates the real parts of the analytic signals of all rhythms. For any time point, the observation $y_t$ is a scalar. The noise for the observation $v_t$ is Gaussian with covariance. $\sigma_R^2$ The model equations are:

$$X_t = [(x_t^1)^T, ..., (x_t^N)^T]$$
$$x_t^j = a_j O(\omega_j) x_{t-1}^j + u_t^j, u_t^j \sim N(0, Q_j)$$
(1)

$$\theta_j^t = arg(x_j^t)$$

$$\theta_j^t = arg(x_j^t)$$
(2)

$$O(\omega_j) = \begin{pmatrix} \cos(\omega_j) & -\sin(\omega_j) \\ \sin(\omega_j) & \cos(\omega_j) \end{pmatrix} \quad Q_j = \begin{pmatrix} \sigma_j^2 & 0 \\ 0 & \sigma_j^2 \end{pmatrix}$$
(3)

$$y_t = MX_t + v_t, v_t \sim N(0, \sigma_R^2)$$
(4)

$$M = [1, 0, 1, 0, ...].$$
(5)

We note that the state ($Q$) and observation ($\sigma_R^2$) covariance matrices are estimated from the data, as described below. While we term $x_j^t$ as tracking the analytic signal, this nomenclature differs from the signal processing literature. In signal processing, the analytic signal is defined as a signal that has zero power at negative frequencies. However, given the stochastic frequency modulation permitted in the state vector $x_j^t$, we cannot guarantee that $x_j^t$ has zero power at negative frequencies. Nevertheless, since for sufficiently complex signals there are (potentially) several ways one could represent the dynamics of the amplitude and phase (*Boashash, 1991*; *Rosenblum et al., 1997*), we use the term *analytic signal* here for ease of understanding our intent with this model and comparing it to classical Hilbert-based methods.

## Real-time phase estimation

We perform real-time estimation of phase as follows. First, we use an existing interval of data to acausally fit the parameters of the state space model $a_j$, $\omega_j$, $Q_j$, $\sigma_R^2$ using an expectation-maximization (EM) algorithm as proposed by *Soulat et al., 2019*; for details, see the supplementary information in *Soulat et al., 2019*. Under the EM approach, optimization of parameters follows a two-step process. First, initial values for parameters are selected, usually from prior knowledge such as an examination of the power spectrum of an initial data sample. This allows the algorithm to estimate the expectations for the state and the observation. Second, using the state and observation estimates, an analytic solution exists (*Shumway and Stoffer, 1982*) for the parameters that maximizes the likelihood. We repeat the expectation and maximization procedures until the parameter estimates do not change between two iterations beyond a threshold. The rate of convergence of the EM algorithm (speed of computation) depends on the initial parameter estimates and the signal-to-noise ratios (SNR) of the rhythms present in the signal.

In simulations, unless otherwise specified, we use the first 2 s of data to fit the model parameters. With these model parameters estimated and fixed, we then apply a Kalman filter to predict and update the state estimates, and estimate the phase and amplitude for each oscillator, each representing a different rhythm, for every sample (*Figure 1*). We note that the Kalman filter is an optimal filter, that is, in the presence of Gaussian noise, the Kalman filter achieves minimum error and is an unbiased estimate of the mean phase. However, if the noise is not Gaussian, the Kalman filter remains the optimal linear filter for phase estimation under this model. Given the stochastic frequency modulation that is possible under the model, the Kalman filter is robust to small shifts in the central frequency of the rhythm.

We now define the stages of Kalman filtering. The state is initialized with zeros and the state covariance to a diagonal matrix with 0.001 along the diagonal. To predict the future state and state

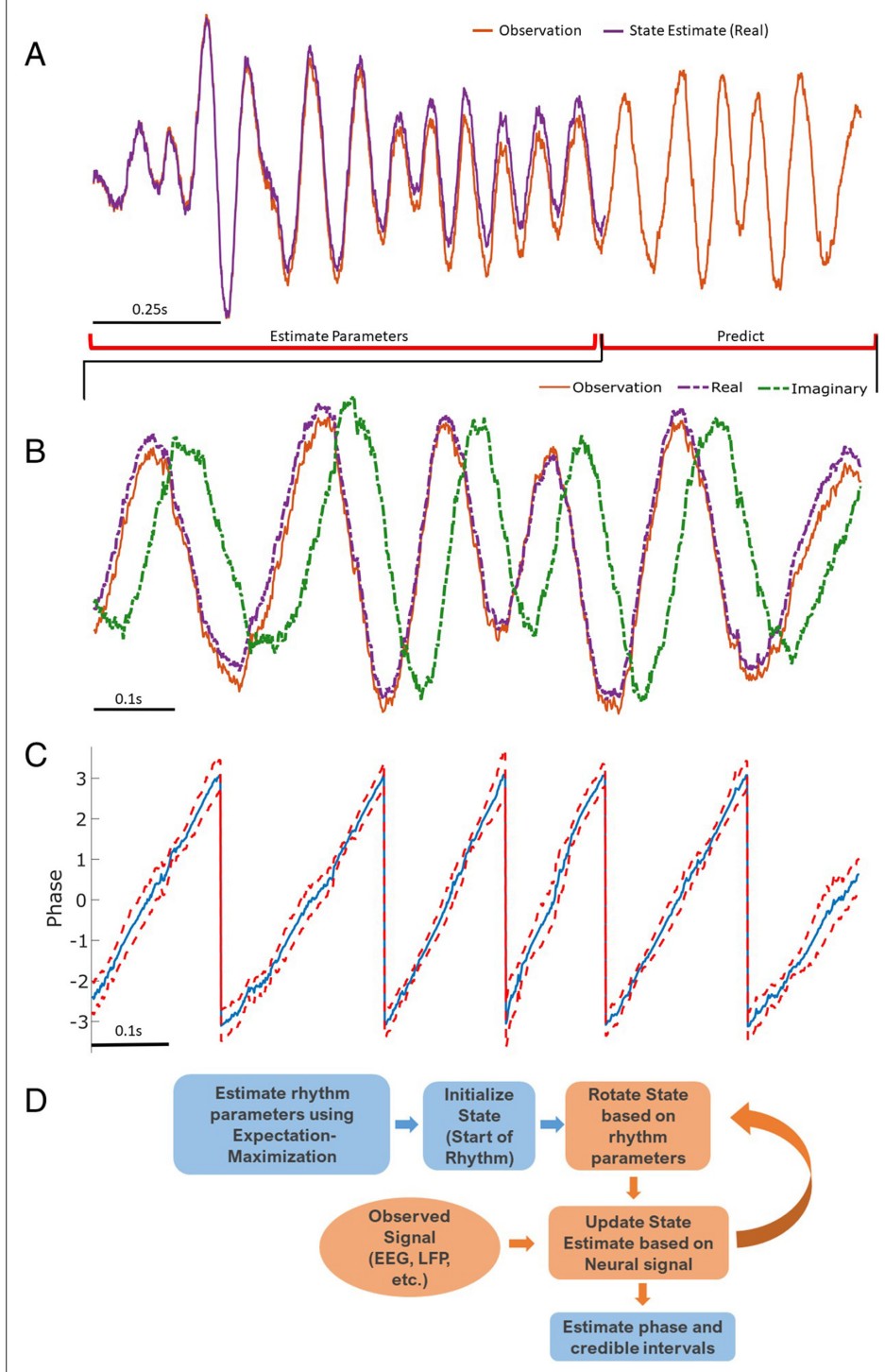

**Figure 1.** Illustration of state space phase estimate (SSPE) procedure. Given an observation (**A**, red), we estimate model parameters in an initial interval of data. At subsequent times, we use the observed data to (**B**) predict and update the state estimate (purple), and (**C**) the phase and credible intervals, causally estimated. Note that prediction can be done for any time after parameter estimation; here we show a representative time interval. (**D**) Conceptual diagram of the steps involved in the SSPE method.

estimation error at time $t$ prior to filtering ( and ) given the filtered state and estimation error ($x_{t-1}^{t-1}$ and $P_{t-1}^{t-1}$) from the previous sample ($t-1$), we compute:

$$x_t^{t-1} = aO(\omega)x_{t-1}^{t-1} \tag{6}$$

$$P_t^{t-1} = aO(\omega_j)P_{t-1}^{t-1}(aO(\omega_j))^T + Q. \tag{7}$$

We then estimate the Kalman gain for the current sample ($t$):

$$K_t = P_t^{t-1}M^T(MP_t^{t-1}M^T + \sigma_R^2)^{-1}. \tag{8}$$

Finally, we filter the current sample using the Kalman gain to compute filtered state ($x_t^t$) and state estimation error ($P_t^t$) at time $t$:

$$x_t^t = x_t^{t-1} + K_t(y_t - Mx_t^{t-1}) \tag{9}$$

$$P_t^t = P_t^{t-1} - K_tMP_t^{t-1}. \tag{10}$$

The SSPE method also tracks the state estimation error ($P_t^t$), which allows estimation of confidence in the mean phase estimates. To do so, we sample 10,000 samples from the posterior distribution of each state $j$, and from this estimate the width of the 95 % credible intervals for the phase. The credible intervals are derived from the posterior distribution of the state, which is Gaussian; however, following their transformation into angles, the intervals are not constrained to be symmetric. Note that, alternatively, one could attempt to define a closed form for the phase distribution to estimate confidence bounds, as we estimate the posterior distribution for the analytic signal under the Kalman filter. However, this strategy requires an expectation of asymptotic convergence and the assumption of independence of amplitude and phase (*Withers and Nadarajah, 2013*).

## AR-based forecasting to estimate phase

While current techniques depend upon filtering to isolate a narrowband rhythm from which to estimate phase, different methods exist to forecast the phase. One category of methods forecasts the phase as a linear extrapolation from the current moment while assuming a singular central frequency, which is estimated from the data (*Rodriguez Rivero and Ditterich, 2021*; *Rutishauser et al., 2013*). Other methods estimate phase by using an AR model. After estimating the AR parameters (either asynchronously or synchronously), these methods forecast data to limit filter edge effects, then use a filter and Hilbert transform to estimate phase (see *Blackwood et al., 2018*, and *Zrenner et al., 2020*). We compare the SSPE to this latter set of techniques as these are more flexible to variability in rhythms than methods that assume a constant central frequency when estimating phase. We implement the version of the *Zrenner et al., 2020*, algorithm accessible at https://github.com/bnplab/phastimate: and term this method as *Zrenner*. Please refer to *Zrenner et al., 2020*, for algorithm details. The Zrenner algorithm uses an FIR filter (applied forward and backward) and the Hilbert transform as the core phase estimation technique. The AR model used to forecast the data and allow real-time phase estimation (by expanding the window for the FIR filter) is fitted to the 30th order and efficiently updated on every new sample. Compared to the default settings, we set the algorithm to track a 6 Hz rhythm by increasing the window size for the FIR filter to 750 ms, the filter order to 192, and the frequency band to 4–8 Hz. We utilize the implementation of the *Blackwood et al., 2018*, algorithm as provided by the authors. In *Blackwood et al., 2018*, the authors develop a forward bandpass filter – AR-based (fifth-order AR model whose parameters are fit every second) forecasting and then the Hilbert transform – to compute the phase. Here, we instead apply a Hilbert transformer (of order 18) that estimates the phase by directly computing the Hilbert transform while filtering the data as implemented in *Blackwood, 2019*. We do so because this implementation is more computationally efficient, functionally equivalent to the original method, and the code is made widely available. We verified in sample data that application of the original and modified Blackwood methods produced consistent phase estimates. We refer to this method as *Blackwood*.

## Non-causal phase estimation

Existing studies compare causal, real-time phase estimation approaches to an acausal estimate of phase computed by applying an (acausal) FIR filter and the Hilbert transform (*Blackwood et al., 2018*; *Chen et al., 2013*; *Zrenner et al., 2020*). To maintain consistency with the existing literature, we do so here and apply a least-squares linear-phase FIR filter of order 750, with bandpass 4–8 Hz. After forward and backward filtering of the data (using MATLAB's *filtfilt* function), we apply the Hilbert transform to compute the analytic signal (using MATLAB's *hilbert* function). We estimate phase using the four-quadrant arctan function implemented by MATLAB's *angle* function and estimate the amplitude envelope using MATLAB's *abs* function. We use the *angle* function throughout the analysis to estimate the phase from a complex valued analytic signal. This acausal estimate of phase has been proposed to serve as a lower bound in phase estimation error for methods that apply a filter-based approach to real-time phase estimation (*Zrenner et al., 2020*).

## Defining error as circular standard deviation

Identifying a common error metric across alternative algorithms to causally estimate phase is crucial for accurate comparison. We follow *Zrenner et al., 2020*, and use the circular standard deviation of the difference between the estimated phase and true phase:

$$CircSD = \sqrt{-2\log(|\exp\left(\frac{1}{n}\sum_{k=1}^{n}(i\theta - i\hat{\theta})\right)|)}$$

(11)

where $n$ is number of samples, $\theta$ is true phase, $\hat{\theta}$ is the phase estimate, and $||$ indicates the absolute value. We transform the final result from radians to degrees. The circular standard deviation (a transformed version of the phase locking value) captures the variance in the circular error for the phase estimate and ignores any potential bias (non-zero mean shift).

## Simulated data

### Narrowband to broadband rhythms

Our initial simulation tests the capability of different real-time phase estimators to track different frequency band profiles. We shift the bandwidth of the target rhythm from narrow (sharply peaked in the spectral domain at 6 Hz) to wide (broadly peaked in the spectral domain; peak bandwidth 4–8 Hz). We simulate all rhythms for 10 s at a sampling rate of 1000 Hz, with a central frequency of 6 Hz. For each simulated rhythm, we apply each phase estimation method, as described in Results.

The first type of rhythm we simulate is a pure sinusoid with added white noise:

$$Y = 10\cos(2\pi * 6 * T) + \epsilon_1$$

$$\epsilon_1 \sim N(0, 1)$$

(12)

where $T$ is time, and $N(0,1)$ indicates a standard normal distribution. The second type of rhythm we simulate is a pure sinusoid with added pink noise, that is, the power spectral density (PSD) of the noise decreases with the reciprocal of frequency ($f$):

$$Y = 10\cos(2\pi * 6 * T) + \epsilon_1$$

$$PSD(\epsilon_1(f)) \sim \frac{1}{f^{1.5}}.$$

(13)

The third type of rhythm we simulate is filtered pink noise with added pink noise. To create a broadband spectral peak, we apply the same FIR filter utilized in the acausal phase estimation analysis (see Materials and methods: Non-causal phase estimation) to pink noise ($PSD \sim 1/f^{1.5}$), normalizing by the standard deviation and amplifying the resulting signal 10-fold. The resulting signal consists of a broad spectral peak centered at 6 Hz. To this broadband signal we again add independent pink noise ($\epsilon_1$) as in *Equation 13*.

For the final rhythm we simulate a signal $y_t$ from the SSPE model (see Materials and methods: State space model framework) which is non-sinusoidal and broadband. The state in the state space model is an AR model of order 2 (*West, 1997*) (equivalently, a damped harmonic oscillator driven by noise), which has been proposed to capture features of gamma rhythms in visual cortex and rhythms in EEG (*Franaszczuk and Blinowska, 1985*; *Xing et al., 2012*; *Burns et al., 2010*; *Spyropoulos et al., 2019*). The addition of observation noise under the state space model can be analogized to measurement

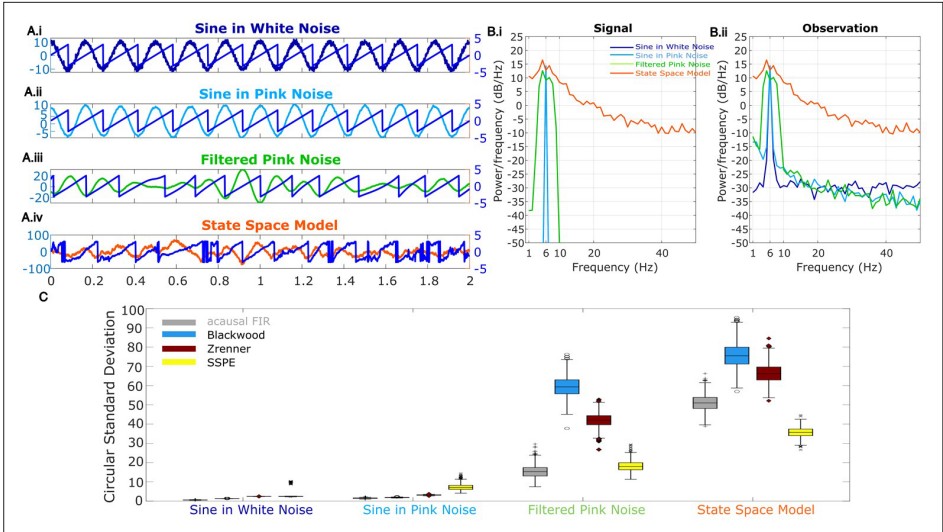

**Figure 2.** Wide band rhythm's phase better tracked with state space phase estimate (SSPE). (**A**) Example 2 s of simulated observed data (thick curves) and the true phase (thin blue curves) for each scenario. (**B**) For each scenario, example spectra of (B.i) the signal, and (B.ii) the observation (i.e., signal plus noise). Spectra were estimated for 10 s segments using the function 'pmtm' in MATLAB, to compute a multitaper estimate with frequency resolution 1 Hz and nine tapers. (**C**) The phase error for each estimation method (see legend) and simulation scenario. In each box plot, the central mark indicates the median; the bottom and top edges of the box indicate the 25th and 75th percentiles, respectively; the whiskers indicate the most extreme data points not considered outliers.

The online version of this article includes the following figure supplement(s) for figure 2:

**Source data 1.** Circular standard deviation for all methods.

noise. Thus, the state space model provides a simple possible model for brain rhythms observed in electrophysiological data. For the model, we fix one latent state ($j = 1$) with frequency 6 Hz ($\omega_j = 6$), and set $a$ (scaling factor) = 0.99, variance $Q = 10$, and observation noise variance ($\sigma_R^2$) to 1. For this rhythm, phase was estimated from the state directly as described in Materials and methods: State space model framework. We show example instances of all four simulated rhythm types in *Figure 2A*.

## Two simultaneous rhythms with nearby frequencies

In many physiologic use cases (*Tort et al., 2010*; *VanRullen, 2016*), we may wish to track one rhythm while ignoring another at a nearby frequency. This may occur because of low spatial resolution (e.g., from mixing of signals due to volume conduction; *Nunez and Srinivasan, 2006*) or because there exist overlapping neural populations expressing different rhythms (*Kocsis et al., 1999*). To simulate this case, we consider two sinusoids. We define the target rhythm to track as a 6 Hz sinusoid with consistent amplitude. We define the confounding rhythm as a sinusoid with varying frequency and amplitude. We define the observation (signal plus noise) as:

$$Y = 25\cos(2\pi * 6 * T) + A * 25\cos(2\pi * F * T + \tfrac{\pi}{4}) + \epsilon_1$$

$$\epsilon_1 \sim N(0, 0.5) \tag{14}$$

where $A \in [0.2, 2]$ and $F \in [1, 11]$. The confounding rhythm frequency assumes a range of integer values from 1 to 11 Hz. We note that replacing the constant phase shift ($\pi/4$) with a randomly selected (uniform $[-\pi, \pi]$) phase shift produces qualitatively consistent results. The confounding rhythm amplitude ranges from 0.2 to 2 times the amplitude of the target rhythm, in increments of 0.2.

## Phase reset

Ongoing slow rhythms in the brain occasionally undergo a sudden shift in phase. This can occur both in response to a stimulus or as a result of ongoing spontaneous neural dynamics (*Voloh and*

*Womelsdorf, 2016*). To simulate sudden changes in a rhythm's phase, we implement a pure sinusoid (6 Hz) with additive pink noise and impose phase slips at four distinct time points, separated by at least 1 s. We fix the extent of phase slip, relative to the previous phase, at $\pi/2$ . However, the phase at which the slip occurs differs across the four time points (see example in Figure 4A). We simulate the signal in five intervals, as follows:

$$Y(0 < T < 3.5) = 10\cos(2\pi * 6 * T) + \epsilon_1 \tag{15}$$

$$Y(3.5 < T < 4.75) = 10\cos(2\pi * 6 * (T - 3.5) + \tfrac{\pi}{2}) + \epsilon_1$$

$$Y(4.75 < T < 6.5) = 10\cos(2\pi * 6 * (T - 4.75)) + \epsilon_1$$

$$Y(6.5 < T < 8.75) = 10\cos(2\pi * 6 * (T - 6.5) + \tfrac{\pi}{2}) + \epsilon_1$$

$$Y(8.75 < T < 10) = 10\cos(2\pi * 6 * (T - 8.75)) + \epsilon_1$$

$$PSD(\epsilon_1(f)) \sim \tfrac{1}{f^{1.5}}$$

### Signal-to-noise ratio

To investigate the impact of SNR on phase estimation, we again simulate a filtered pink noise signal (centered at 6 Hz) with added pink noise (the third type of rhythm simulated in Materials and methods: Narrowband to broadband rhythms). To modulate the SNR, we first compute:

$$SNR_0 = \sigma_{signal}/\sigma_{noise}$$

where $\sigma_{signal}$ ($\sigma_{noise}$) is the standard deviation of the filtered pink noise signal (added pink noise). We then divide the filtered pink noise signal by $SNR_0$, multiply the filtered pink noise signal by the chosen SNR (1, 2.5, 5, 7.5, 10), and add this scaled signal to the pink noise.

### In vivo data

We consider two datasets to demonstrate how the SSPE method performs in vivo. The first dataset consists of LFP activity collected from the infralimbic cortex in rats. LFP were acquired continuously at 30 kHz (Open Ephys, Cambridge, MA). An adaptor connected the recording head stage (RHD 2132, Intan Technologies LLC, Los Angeles, CA) to two Mill-max male-male connectors (eight channels each, Part number: ED90267-ND, Digi-Key Electronics, Thief River Falls, MN). For additional details on data collection, please refer to *Lo et al., 2020*. We analyze here 250 s of LFP data from one channel. Before real-time phase estimation, we downsample the data to 1000 Hz (with a low pass filter at 100 Hz to prevent aliasing) for computational speed. To model these data, we fit three oscillators with the SSPE method, with the goal of tracking a target theta rhythm (defined as 4–11 Hz). Motivated by an initial spectral analysis of these data, we choose one oscillator to capture low-frequency delta band activity ($\omega_j$ = 1 Hz, based on spectral peak), one to capture the theta band rhythm of interest ($\omega_j$ = 7 Hz), and a final oscillator to track high-frequency activity ($\omega_j$ = 40 Hz). We estimate these model parameters (so center frequencies may change) using the first 10 s of LFP data, and then fix these model parameters to estimate future phase values (up to 250 s). In doing so, we assume that the same oscillator models in the SSPE method remain appropriate for the entire duration of the LFP recording.

The second dataset was derived from the publicly available data in *Zrenner et al., 2020*. The data consist of spatially filtered and downsampled scalp EEG recordings from electrode C3 (over central areas), which records a strong somatosensory mu rhythm (8–13 Hz). For details on data collection and preprocessing, see *Zrenner et al., 2020*. We analyze 250 s of data from a single participant for demonstration purposes. For these data, before the real-time phase estimation step, we first apply a moving average causal filter (using 3 s of past data for each new data point) to remove slow drifts. We again track three oscillators in the SSPE method, with the goal of estimating phase from the mu rhythm: one in the delta band ($\omega_j$ = 2 Hz), one centered at the mu rhythm ($\omega_j$ = 10 Hz), and one in the beta band ($\omega_j$ = 22 Hz). We note that these frequency values initialize the model and are then tracked

in time. In analyzing the mu rhythm, we ensured that phase was estimated from an oscillator whose frequency was in the mu rhythm range, that is, 8–12 Hz.

For the in vivo data, the true phase is unknown. To estimate phase error, we define the true phase as follows: first, we select a 10 s segment centered on the 1 s interval of the phase estimate. We then fit the state space model (*Equations 1-4*) acausally to this 10 s segment and estimate the phase in the 1 s interval at the segment's center. We note that here, the phase estimate is acausal because we use forward and backward smoothing of the state estimate when estimating the analytic signal. We also note that, in some segments, the frequency band of interest may not be identified through EM due to low signal strength. When this occurred, we omit error in the phase estimates, as these intervals lack the frequency of interest.

### Real-time implementation in TORTE

To make SSPE more broadly available, we implemented SSPE into the real-time TORTE (Toolkit for Oscillatory Real-time Tracking and Estimation) system (https://github.com/tne-lab/phase-calculator, 10.5281/zenodo.2633295). TORTE is built upon the Open Ephys platform (https://open-ephys.org/) and can be used with a broad range of electrophysiologic acquisition devices. Phase outputs and triggers from the system can be used to drive presentation of psychophysical stimuli or delivery of optical/electrical/magnetic stimulation. We directly integrated the MATLAB-based SSPE algorithm described above into C++ for improved efficiency. TORTE uses a buffer-based approach to receive data, based on the buffered approach of the underlying Open Ephys system. New buffers are repeatedly received by TORTE containing the most recent neural data. These buffers are subject to system latency causing a delay between activity in the brain being received by the toolkit. Both the buffer size and latency individually vary from microseconds to a few milliseconds depending on the acquisition system used. The phase of each sample within the buffer is estimated using SSPE. A detector continuously monitors the estimated phase and can trigger output logic at any desired phase. We note that the observation noise parameter ($Q$) is defaulted to 50, but should be updated by the user to the square root of the amplitude. Two modifications were needed to integrate SSPE into TORTE. For computational efficiency, data were downsampled before application of the SSPE algorithm, and then the resulting phase estimates were upsampled back to the native sampling rate. The Open Ephys system can provide buffer sizes that are not exactly divisible into downsampled buffers. When this occurs, the phase estimates at the end of the upsampled buffer cannot be generated from the downsampled SSPE result. To account for this, an additional predicted state sample (with credible intervals) is computed by forward filtering using *Equations 07; 08*. This extra value is inserted at the end of the downsampled buffer. This provides an extra data point in the SSPE result that can be used to generate the remainder of samples for the upsampled phase output buffer. The second modification results from the real-time data acquisition by TORTE. In this case, low pass filtering must be causal, not the acausal (i.e., forward-backward filtering) used in the MATLAB implementation. To account for the consistent phase shift introduced in a real-time system, TORTE includes an online learning algorithm that adjusts target phase thresholding to improve event timings in closed loop applications. We test the real-time implementation on an AMD FX-8350 CPU with 16 GB RAM.

### Code availability

The code to perform this analysis is available for reuse and further development at MATLAB (https://github.com/Eden-Kramer-Lab/SSPE-paper, copy archived at swh:1:rev:6d80fe9c5f610d0dfffe-23b5eef2012f780ba621, *Mark, 2021b*) and Open Ephys (https://github.com/tne-lab/phase-calculator, copy archived at swh:1:rev:e11421f23403399ca6b2d85132a31e2c1d5b1397, *Mark, 2021a*).

## Results

We first examine in simulations the viability of different causal phase estimation algorithms, and how these existing methods compare to the proposed method – the SSPE. In these simulations, we vary the nature of the underlying rhythm of interest – either the spectral bandwidth or the noise/confounding signal present. Through these simulations, we examine the assumptions under which different methods perform best at estimating the phase in real-time. We then illustrate the application of the SSPE method to in vivo data in two case studies: a rodent LFP and a human EEG.

## SSPE accurately tracks the phase of a single oscillation

To test the accuracy of causal estimates of phase, we simulate rhythms with different spectral profiles in four different ways. For the first two rhythms, we simulate a 6 Hz sinusoid and add either white noise (constant power spectrum, 'sine in white noise', *Figure 2A*.i) or pink noise (power spectrum decreases with the reciprocal of frequency, 'sine in pink noise', *Figure 2A*.ii). For the next simulation scenario, to generate a broadband spectral peak, we filter pink noise into the band of interest using an FIR filter (4–8 Hz, 'filtered pink noise', see Materials and methods), amplify this signal, and then add additional (unfiltered) pink noise (*Figure 2A*.iii). For the last simulation, to generate the signal of interest we sample data from the model underlying the SSPE method. The state space harmonic oscillator model ('state space model') consists of a damped harmonic oscillator driven by noise (the signal) with additive white noise (*Figure 2A*.iv). We note that, for the first two simulation approaches in which the signal is a pure sinusoid, the true phase is known. For the 'filtered pink noise' simulation, we estimate the phase by applying the Hilbert transform to the signal before adding noise (see Materials and methods). For the 'state space model' simulation, we determine phase directly from the model (see Materials and methods). Thus, in all simulation scenarios, we have an independent measure of the true phase.

We repeat each simulation 1000 times with different noise instantiations and calculate error as the circular variance of the difference between the estimated phase and true phase (see Materials and methods). We find that for rhythms with narrowband spectral peaks (sine in white noise and sine in pink noise, *Figure 2B and C*), all estimators perform well, and those estimators that presume a narrowband signal (i.e., the filter-based approaches) perform best. However, for rhythms with broad spectral peaks (filtered pink noise and state space model, *Figure 2B and C*), the SSPE method outperforms the other causal approaches. Moreover, the SSPE method performs as well as (filtered pink noise), or better than (state space model), the acausal FIR method. In what follows, we further explore the implications of this result and the impact on assessment of phase estimator performance. We conclude that the SSPE method performs well in all four simulation scenarios and outperforms existing causal phase estimation methods in two instances with broad spectral peaks.

## SSPE accurately tracks the phase of two simultaneous oscillations

To replicate a situation that has been documented in EEG (e.g., multiple alpha rhythms coexisting) (*VanRullen, 2016*) and in LFP (e.g., multiple theta/alpha rhythms in hippocampus) (*Tort et al., 2010*), we simulate two rhythms with similar frequencies. In this situation, phase estimation requires an approach to identify the presence of two rhythms and target the rhythm of interest. This simulation examines the impact of choosing a specific, previously specified, frequency response to isolate the rhythm of interest, as required by the two comparative causal phase estimation methods implemented here. In some cases, a finely tuned bandpass filter may separate the two rhythms, although such a filter requires sufficient prior knowledge and may be difficult to construct, particularly if the center frequencies are not perfectly stationary. The SSPE method provides an alternative approach to identify the two (or potentially more) rhythms without filtering and then allows tracking of the appropriate target rhythm. Further, while we focus on the phase of the target rhythm here, we note that the SSPE method provides estimates of the amplitude and phase of all modeled rhythms.

To illustrate this scenario, we simulate two slow rhythms at different frequencies. We set the target rhythm as a 6 Hz sinusoid with fixed amplitude, and the confounding rhythm as a 5 Hz sinusoid with 1.5 times the 6 Hz amplitude (*Figure 3A*). The observed signal consists of the combined target and confounding rhythms, plus white noise, simulated for 15 s. We apply each phase estimation method to the observed signal, and estimate the phase of the target rhythm after fitting parameters with 5 s of initial data. Visual inspection for this example (*Figure 3B*) suggests that only the SSPE method accurately tracks the target phase.

Repeating this analysis with the confounding amplitude and frequency varied across a range of values (frequencies 1–11 Hz and amplitudes 20–200% of the target rhythm; see Materials and methods), we find consistent results. For all methods, the error increases as the frequency of the confounding rhythm approaches the frequency of the target rhythm (6 Hz, *Figure 3C*). However, the increased error is restricted to a narrower frequency interval for the SSPE method. For the existing causal phase estimators, the error likely reflects choices in the filtering procedure (i.e., type of filter, features of passband and stopband, window size, see *Sameni and Seraj, 2017*) and modeling procedure (i.e.,

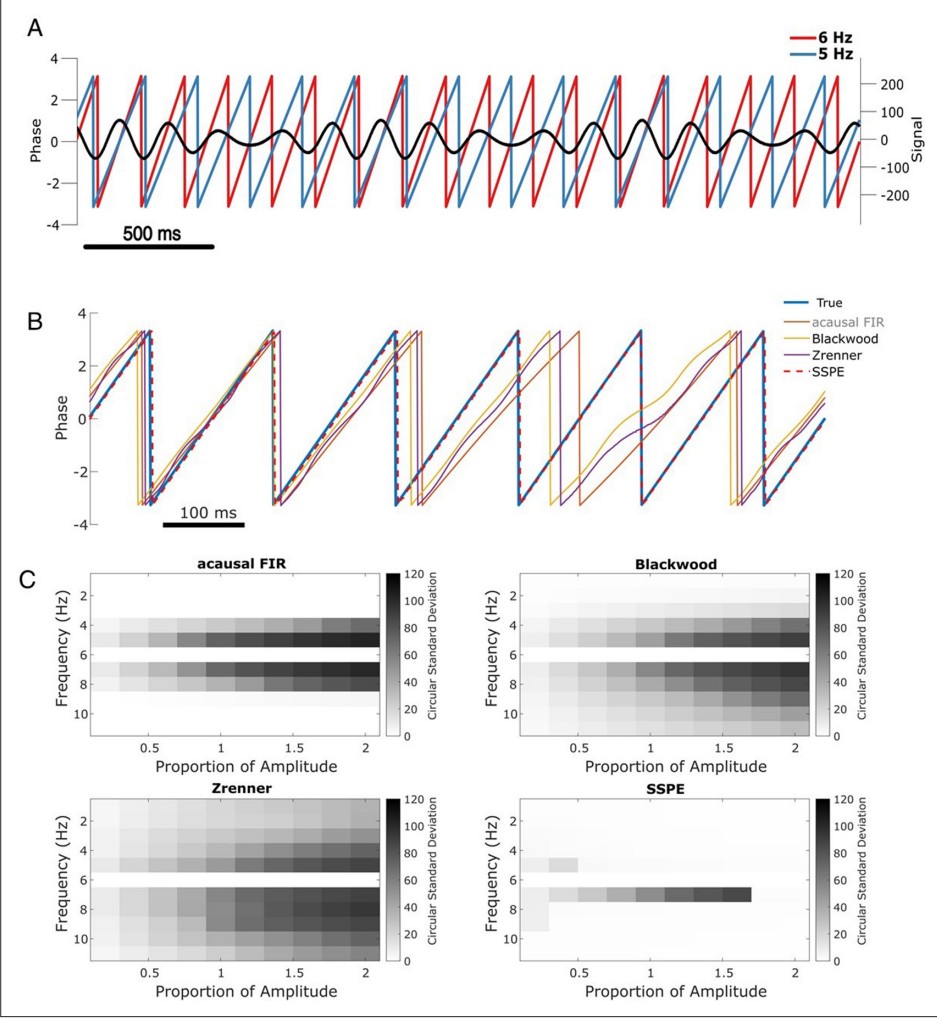

**Figure 3.** SSPE performs well when two oscillations exist at nearby frequencies. (**A**) Example observed signal (black) and phase for two simulated rhythms, the confounding rhythm at 5 Hz (blue) and the target rhythm at 6 Hz (red). (**B**) Example phase estimates for the target rhythm in (**A**) using four different approaches (see legend). (**C**) Estimation error (circular standard deviation, see scale bars) for each estimation method as a function of frequency and amplitude of the confounding oscillation. In all cases the error increases as the frequency of the confounding oscillation approaches 6 Hz (the frequency of the target oscillation), and as the amplitude of the confounding oscillation increases.

The online version of this article includes the following figure supplement(s) for figure 3:

**Source data 1.** Circular standard deviation for all methods on two simultaneous oscillations simulation.

AR order, number of samples forecasted; *Chen et al., 2013*; *Zrenner et al., 2020*). Similarly, for the acausal FIR estimator, the error likely reflects the filter passband of 4–8 Hz, with increasing error as the confounding rhythm nears the 6 Hz target rhythm. For the existing causal and acausal methods, error also increases as the confounding oscillation amplitude increases (*Figure 3C*).

Previously, we showed that all estimation methods performed well for a signal consisting of a single sinusoid in white noise (see *Figure 2*, sine in white noise). Here, we show that for a signal consisting of two sinusoids in white noise, the SSPE method outperforms the existing methods. Unlike these existing methods, the SSPE method estimates parameters for both the target and confounding rhythms, and tracks each oscillation independently. By doing so, the SSPE method accounts for the confounding effect of the nearby oscillation and allows more accurate tracking of the target rhythm. We conclude that the SSPE method outperforms these existing methods in the case of two simultaneous rhythms with nearby frequencies.

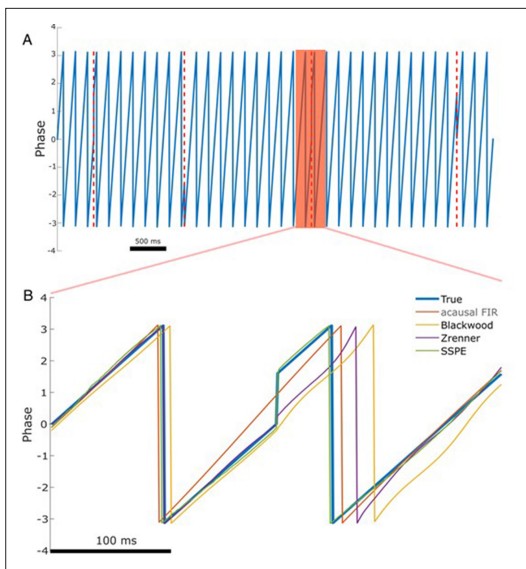

**Figure 4.** The state space phase estimate (SSPE) accurately tracks phase following a phase reset. (**A**) Example phase of a 6 Hz sinusoid (blue) with four phase resets (red dashed lines). (**B**) Example phase estimates (thin curves) and the true phase (thick blue curve) at the indicated phase reset in (**A**). The SSPE method (in green) tracks the true phase much more closely than other methods following a phase reset.

## Phase estimation following phase reset

Phase resets are defined as consecutive samples of data that show a sudden shift in the oscillatory phase, proposed to represent the presence of synchronization of a neural mass in response to a stimulus, for example (*Lakatos et al., 2009*). Phase resets are common and well documented in electrophysiologic recordings (*Fiebelkorn et al., 2011*; *Makeig et al., 2004*). Recent work delivering electrical stimulation based on phase of the theta rhythm has been forced to avoid segments with phase resets to reduce error in real-time phase estimation (*Gordon et al., 2021*). Further, accurate tracking of phase resets would provide better evidence for the relevance of phase resetting in generating event related potentials (*Sauseng et al., 2007*) – an analysis that could also be performed acausally. We examine the performance of each phase estimation method in the presence of simulated phase resets.

To do so, we simulate phase reset as a shift in the phase of a 6 Hz sinusoid by 90 degrees at four points in a 10 s signal (example in *Figure 4A*). We then compute the error of each phase estimation method in the 167 ms (one period of a 6 Hz rhythm) following the phase reset. Visual inspection of an example phase reset (*Figure 4B*) suggests that only the SSPE method accurately tracks the phase in the interval following the reset. We note that we fit SSPE parameters using an initial 2 s interval containing no phase resets (*Equation 15*). Repeating this analysis for 1000 simulated signals (4000 total phase resets), we find consistent results; the error of the SSPE method is significantly lower than the error of the other methods (p ~ 0 in all cases using two-sided t-test, see *Table 1*). These results demonstrate that the existing causal and acausal methods implemented here are unable to estimate the phase accurately immediately after a phase reset. To determine how quickly each method returned to accurate phase estimation, we first estimated the pre-stimulus average error for each method, prior to the first phase reset, over a 500 ms interval. We then determined the time between each subsequent phase reset and the moment when the error reduced to 1.5 times the pre-stimulus error. Note that the level of pre-stimulus error differs across methods, so, for example, we determine when the SSPE converges to a lower error than the Blackwood method. We find that each method eventually converged to accurate phase estimation after a phase reset, with the SSPE method converging most rapidly (*Table 1*). The differences between methods likely result from the windowing used by each method when estimating the phase; smaller

**Table 1.** Error following phase reset for different phase estimation methods.
The state space phase estimate (SSPE) method tracks phase resets with error near 0, and rapidly converges to accurate phase estimation following a phase reset.

|  | SSPE | Blackwood | Zrenner | Acausal FIR |
|---|---|---|---|---|
| Error (std. dev.) | 2.85 (0.89)* | 62.08 (0.38) | 44.68 (0.38) | 15.04 (0.23) |
| Time to convergence (std. dev.) | *34 ms (162)* | *190 ms (33)* | *747 ms (22)* | *555 ms (206)* |

*p~0 compared to all other methods.

The online version of this article includes the following source data for table 1:

**Source data 1.** Circular standard deviation and convergence time information for phase reset simulation.

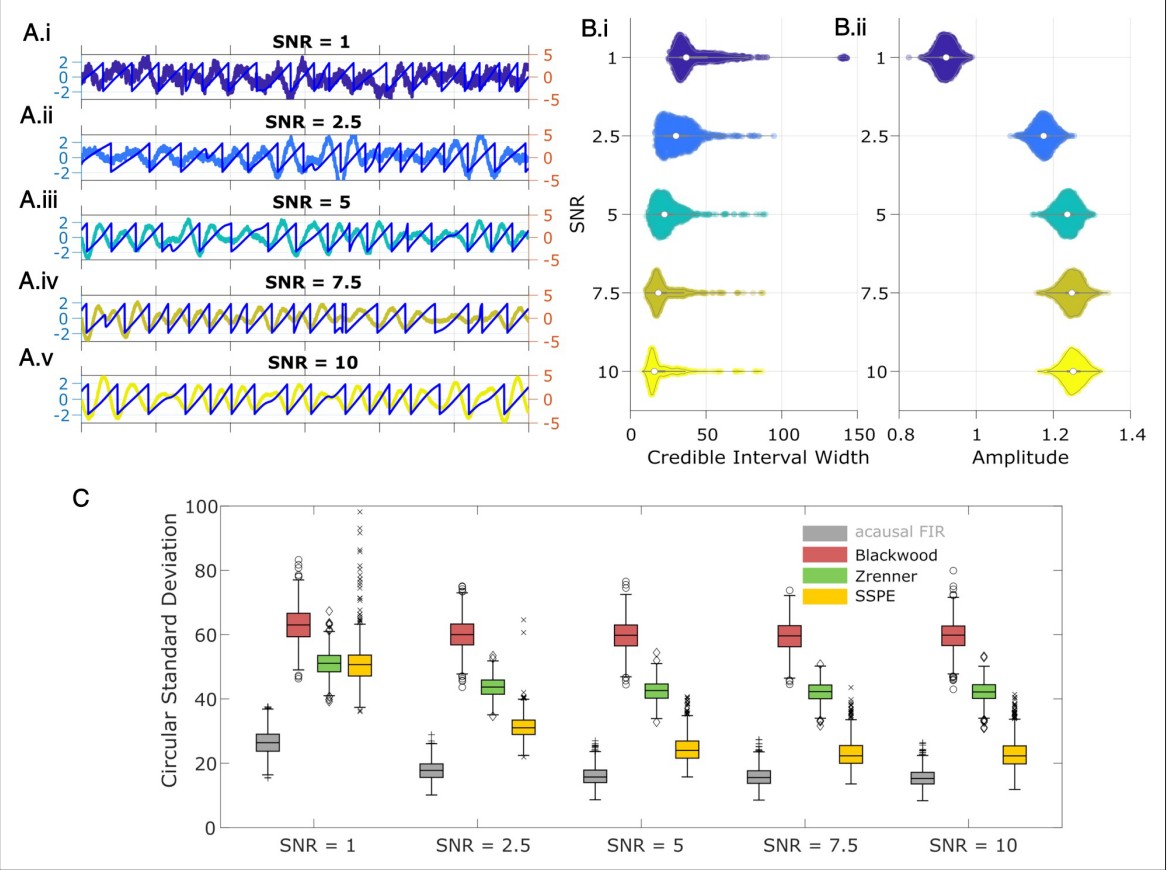

**Figure 5.** State space phase estimate (SSPE) tracks an oscillatory signal with less error across a range of signal-to-noise ratios (SNR). (**A**) Example 2 s of simulated observed data (thick curves) and true phase (thin blue curves) for each SNR value. (**B**) For each SNR, (i) the average credible interval width (colored circles) of the SSPE method for all simulation iterations, and (ii) the average amplitude of the estimated signal (colored circles) for all simulation iterations. White circles (horizontal gray lines) indicate median (25th and 75th percentiles) values for each SNR. (**C**) The phase error for each estimation method (see legend) and simulation SNR. In each box plot, the central mark indicates the median; the bottom and top edges of the box indicate the 25th and 75th percentiles, respectively; the whiskers indicate the most extreme data points not considered outliers.

The online version of this article includes the following figure supplement(s) for figure 5:

**Source data 1.** Circular standard deviation for all methods on signal-to-noise ratio (SNR) simulation.

**Figure supplement 1.** Example power spectra for all simulated cases.

windows will lead to less impact of the phase reset on the error (as seen with the Blackwood approach converging more quickly than the Zrenner method), with the SSPE needing a window of a single sample.

We note that these existing methods all require a bandpass filtering step, which limits the fastest rate at which the phase can change (instantaneous frequency **Boashash, 1991**). The SSPE method, however, does not require the phase change within a limited range of instantaneous frequencies and therefore can track rapid changes in phase. We conclude that the SSPE method is able to most rapidly track the true phase after a phase reset and thus provides the most accurate estimate.

## Phase estimation with different SNR

To demonstrate the impact of noise on phase estimation, we simulate signals with different SNR. To generate a signal we filter pink noise into the band of interest (see Materials and methods: Signal-to-noise ratio), and then add additional (unfiltered) pink noise. The ratio of the standard deviations of the signal and noise defines the SNR; we modulate the standard deviation of the signal to achieve the desired SNR. Finally, we normalize the full observation so that amplitudes across simulations are consistent. In these simulations the SNR varies from 1 (signal and noise have the same total power) to 10 (signal has 100 times the total power of the noise); see example traces of the observation under

each SNR in *Figure 5A*.i–iv, and see example spectra under each SNR in *Figure 5—figure supplement 1*. We repeat each simulation 1000 times with different noise instantiations for each SNR.

We compute two metrics to characterize the impact of signal-to-noise on phase estimation. First, we consider the credible intervals estimated by the SSPE, here measured as the average credible interval width throughout a simulation. As expected, the credible intervals track with the SNR (*Figure 5B*.i); signals with higher SNR correspond to smaller credible intervals. Computing the average amplitude (as derived from the acausal FIR, see *Methods: Non-Causal Phase Estimation*) throughout each simulation, we find no overlap between the distributions of amplitudes at SNR = 1 and SNR = 2.5 (*Figure 5. ii*), while the distributions of credible intervals overlap. This suggests that, at low SNR, an arbitrary amplitude threshold (e.g., amplitude >1) would eliminate all data at SNR = 1 and preserve all data at SNR = 2.5. Alternatively, using the credible interval width allows identification of transient periods of confidence (e.g., credible interval width <40 degrees) at both SNR values. We also note that the credible interval width – in units of degrees – provides a more intuitive threshold than the amplitude, whose intrinsic meaning at any particular value is only understood relatively. These results suggest that, in this simulation study, amplitude serves as a less accurate criterion for confidence in phase compared to the credible interval width.

Second, we consider the error in phase estimates using four different phase estimation methods (*Figure 5C*). We find that the error for the SSPE method reduces more quickly than for the other methods, and at an SNR of 1, the SSPE method is comparable to or better than the contemporary methods of real-time phase estimation. At SNR = 10, the error in the SSPE phase estimation is comparable to the acausal FIR, while the alternative real-time algorithms continue to perform with error similar to an SNR of 1. These results suggest that, for the two existing real-time phase estimation methods, broadband rhythms are difficult to track even with high SNR, consistent with *Figure 2C*.

## Example in vivo application: rodent LFP

Having confirmed the SSPE method performs well in simulation, we now illustrate its performance in example in vivo recordings. We first apply the SSPE method to an example LFP recording from rodent infralimbic cortex (see Materials and methods), which contains frequent intervals of theta band (4–8 Hz) activity. Unlike the simulated data considered above, in the in vivo signal, the target rhythm changes in time (e.g., time intervals exist with, and without, the theta rhythm). Accurate application of the SSPE method requires accurate estimation of the parameters defining the oscillators in the SSPE model. In the simulated data, the stationary target rhythm allowed consistent SSPE model parameter estimation from (any) single interval in time; in those simulations, we estimated the SSPE model parameters from a single, early time interval (see Materials and methods) and then used those (fixed) model parameters for future causal phase estimates.

For the in vivo data, in which the target rhythm may change in time, model estimates from an initial time may be inappropriate – and produce inaccurate phase estimates – at a later time. However, comparing the estimated phase to the true phase (see Materials and methods), we find that the phase error tends to remain consistent in time (blue curve in *Figure 6A*; circular standard deviation 95% interval = [33.11, 75.39]; for error of phase estimates computed using alternative methods, see *Figure 6—figure supplement 1*). We find no evidence of a linear trend in error with time (linear fit, slope = 0.01, p = 0.53); that is, we find no evidence that the error increases in time. Instead, visual inspection suggests that the error increases when the spectral content of the LFP changes (e.g., near 120 s in *Figure 6B*). We note that the error bounds are within the bounds of the range expected for real-time phase estimates in *Zrenner et al., 2020*. Finally, we note that visual inspection of the PSD (*Figure 6C*) demonstrates that the theta band power appears more closely matched by a broadband peak, rather than a narrowband peak.

An alternative approach is to re-estimate the SSPE model parameters over time. By allowing the parameters of the model oscillators to change in time, this approach may improve the accuracy of the phase estimates. To examine this in the in vivo data, we refit the SSPE model parameters in each 10 s interval of the LFP data. More specifically, we refit model parameters in every 10 s interval of the 250 s recording, incrementing the 10 s interval by 1 s, to create 240 models. We then use each of these models to causally estimate the phase for all future times. For example, we use the SSPE model with parameters estimated from the interval 40–50 s to then estimate the phase for data after 50 s. We compute the error in the phase estimates for all 240 models to assess whether refitting the

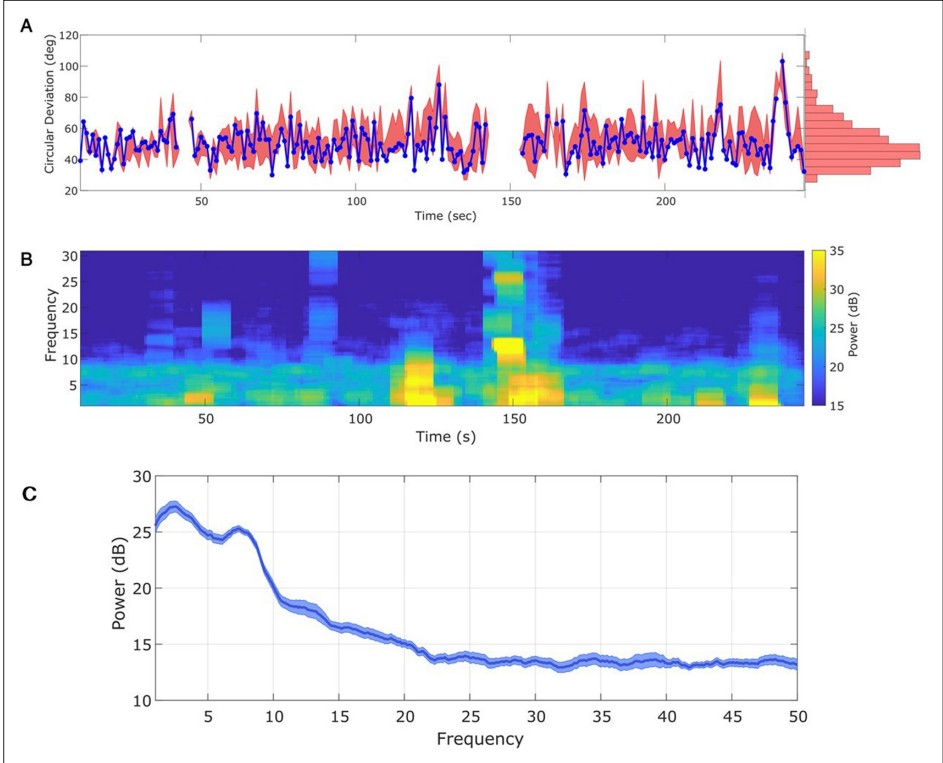

**Figure 6.** Any interval with a prominent rhythm of interest (here: theta) can be used to fit state space phase estimate (SSPE) parameters. (**A**) The phase error for a single instance of the model fit (using data at times 10–20 s, blue curve), and the 90% interval for phase error (red bands) at time *t* derived using parameter estimates from all models estimated with data prior to *t*. The moments where no error is reported are moments when an acausal approach failed to detect the theta band rhythm. Right: a histogram of circular deviation across all time demonstrating that error remains below 60 degrees in the majority of cases. (**B**) Corresponding spectrogram of the rodent local field potential (LFP) data (multitaper method with window size 10 s, window overlap 9 s, frequency resolution 2 Hz, and 19 tapers). (**C**) Average power spectral density (PSD) of (**B**) over all times with theta band oscillator identified in (**A**). The mean PSD (dark blue) and 95 % confidence intervals (light blue) reveal a broadband peak in the theta band.

The online version of this article includes the following figure supplement(s) for figure 6:

**Source data 1.** Error and spectrogram for in vivo local field potential (LFP) data.

**Figure supplement 1.** The state space phase estimate (SSPE) method performs optimally across algorithms when tracking local field potential (LFP) theta.

SSPE model parameters impacts future phase estimates. We find no evidence that the distribution of phase errors changes for the refit models compared to the original fixed parameter model for the immediate 30 s that followed (p = 0.21, Wilcoxon rank sum test) or for the entire time period analyzed (red interval in *Figure 6A*, circular standard deviation 95% interval = [30.68, 88.7] degrees; p = 0.35 using Wilcoxon rank sum test). The original fixed parameter SSPE model (fit using the first 10 s of data; blue curve in *Figure 6A*) performs as well as the models refit with future data. We conclude that, under the assumption that parameters for the frequency band of interest are stable, we are able to estimate the SSPE model parameters whenever there exists a consistent rhythm of interest and then apply this model to estimate phase at future times (here on the scale of hundreds of seconds). In other words, for these data, any 10 s interval with sufficiently strong theta rhythm can be used to fit parameters before the model is applied causally. This result has an important practical implication; the SSPE model parameters can be fit using an initial segment of data, and then applied to derive causal estimates of future phase values. In other words, the model has robust estimation properties; parameters estimated early in the recording remain appropriate for future observations.

In addition, these results allow a novel insight into the rhythmic activity of the data, beyond those available using traditional methods. Typical analysis of neural rhythms either presumes the presence of

a rhythm (for subsequent filtering) or operates in the frequency domain using the PSD and statistical testing to identify significant rhythms. Here, through direct fitting of the state space model to the time series data, we find consistent model parameters that accurately track the observed theta rhythm. This consistency is maintained through variations in the rhythm's amplitude. These observations suggest that the theta rhythm in infralimbic LFP maintains a consistent underlying rhythmic structure over the entire duration of the observation.

## Confidence in the phase estimate

Past approaches to causally estimate phase have not attempted to quantify confidence in the phase estimate beyond utilizing the amplitude as a surrogate (*Zrenner et al., 2020*). However, using the amplitude assumes a monotonic linear relationship between the phase confidence bounds and the amplitude, which may not be true (e.g., for an AR(2) process with complex roots, *Spyropoulos et al., 2019*). Unlike existing methods, the SSPE method provides real-time estimates of the credible interval estimates for the phase (resulting from estimating the parameters of the SSPE model and tracking the phase using the Kalman filter; see Materials and methods). To illustrate this, we show the phase and credible interval estimates of the SSPE method for an example 3 s interval of the LFP data (*Figure 7*; for comparison to phase estimates computed using alternative methods, see *Figure 7—figure supplement 1*). We note that, during a prominent theta rhythm, the credible intervals tightly track the mean phase estimate; at other times – when the theta rhythm is less obvious – the credible intervals expand. We can restrict error in the phase estimate by examining only samples with small CI width. When thresholding at 10 degrees of credible interval width, we find that the error decreases from (mean) 46.9 (s.e. 0.89) degrees to 26.91 (s.e. 1.07) degrees, while still retaining 27% of the data. This is within the current state-of-the-art in phase estimation with minimum of 20–40 degrees error as a function of SNR (when applying a sufficiently high amplitude threshold, *Zrenner et al., 2020*).

Previous studies have proposed a relationship between the theta amplitude envelope, estimated using the traditional approach of narrowband filtering followed by a Hilbert transform, and confidence in the phase (*Zrenner et al., 2020*); when the theta signal is strong (i.e., the amplitude is large), the theta phase estimate improves. To examine this relationship here, we plot the theta amplitude envelope versus the phase credible intervals derived from the SSPE method (*Figure 7C.i*). Visual inspection reveals that, when the amplitude is large, small decreases in amplitude coincide with increased credible intervals, as expected. However, if we choose to approximate certainty in the phase estimate through thresholding the amplitude, we find that fixed amplitude thresholds produce wide ranges of credible intervals (*Figure 7C.ii*). This is especially apparent at lower amplitude thresholds; choosing a threshold of 65 % results in credible intervals that range from 3.8 (2.5th percentile) to 96.8 (97.5th percentile) degrees. Therefore, a fixed amplitude threshold does not guarantee an accurate phase estimate. Instead, we propose that inspection of the credible intervals better guides certainty in the phase estimate. These credible intervals are directly derived from the modeling procedure, as opposed to surrogate measures of confidence in the phase estimate, such as amplitude. In addition, credible intervals are directly interpretable unlike amplitude thresholds, which depend on the measurement scale (e.g., a credible interval threshold of 10 degrees) versus an amplitude threshold of 10 mV. We conclude that amplitude is linked to confidence in the phase, however, a fixed amplitude threshold is not directly interpretable and encompasses a wide range of credible interval values.

## Example in vivo application: human EEG

To illustrate application of the SSPE method in another data modality, we analyze an example EEG recording of the mu rhythm (8–13 Hz) from *Zrenner et al., 2020*. To do so, we apply the SSPE method to track three oscillators (1, 10, 22 Hz) in 10 s intervals (see Materials and methods), and examine the phase estimates of the 8–12 Hz oscillator. Consistent with the rodent LFP data, we find that refitting of the SSPE model parameters is not required. Instead, the SSPE model parameters estimated from an initial 10 s segment of the EEG produce phase estimation errors consistent with SSPE models refit at future times (*Figure 8A*). Like the theta rhythm in the LFP rodent data, the mu rhythm in the human EEG data maintains a consistent spectral profile in time (*Figure 8B*) and further, it appears to be narrowband (*Figure 8C*). Therefore, estimating SSPE model parameters early in the data accurately captures the mu rhythm tracked later in the data, and there is no need to perform the (computationally expensive) refitting of the model.

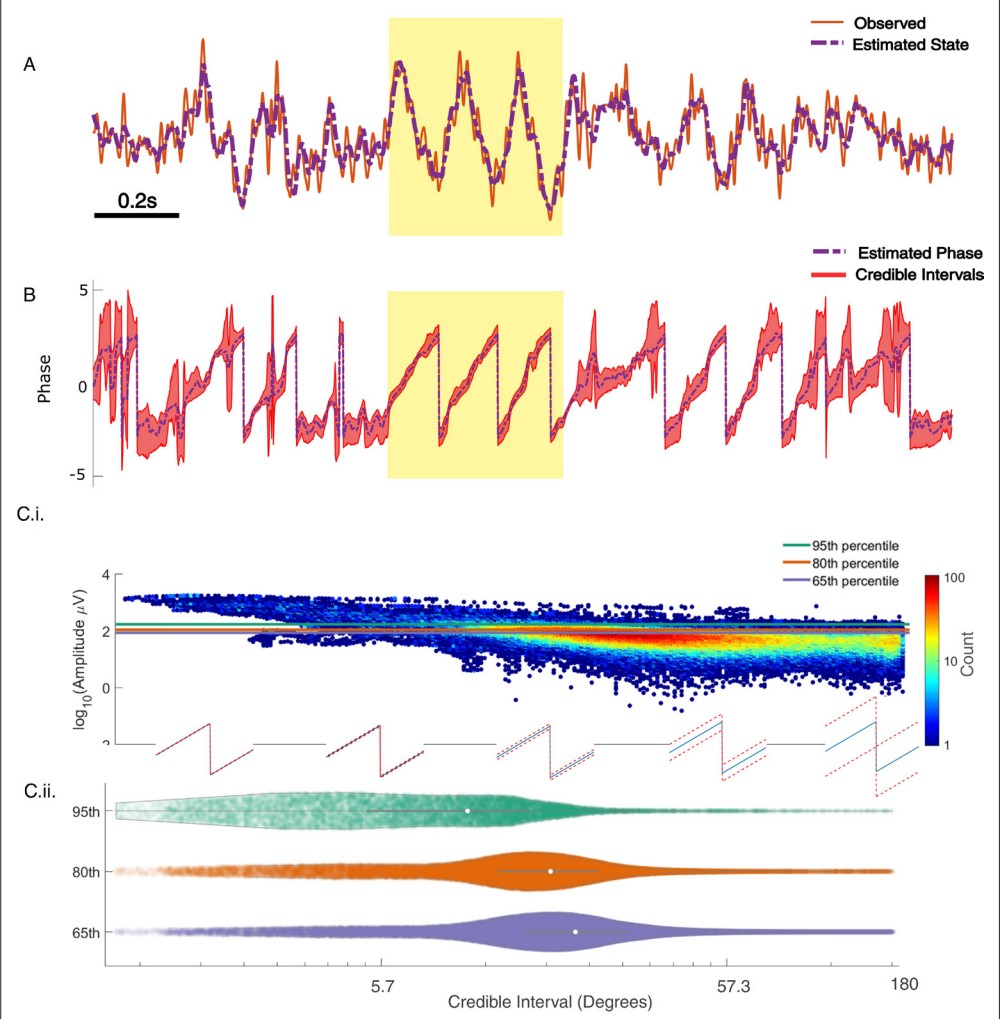

**Figure 7.** State space phase estimate (SSPE) tracks in vivo credible intervals for the phase. (**A**) Example rodent local field potential (LFP) data (red, solid) with a consistent broadband peak in the theta band (4–8 Hz). The estimated state of the SSPE method (purple, dashed) tracks the observed LFP. (**B**) Phase estimates (purple, dashed) and credible intervals (red, shaded) for the example LFP data in (**A**). When the rhythm appears (yellow shaded region) the credible intervals approach the mean phase; when the rhythm drops out, the credible intervals expand. (C.i) Phase credible intervals versus theta rhythm amplitude for each time point shown as a two-dimensional histogram (color indicates count of time points; see colorbar). (Lower) Example cycles of a rhythm with credible intervals, see x-axis of (C.ii) for numerical values of credible intervals. (C.ii) Violin plot of credible intervals for thresholds set at the 65th, 80th, and 95th percentile of the distribution of amplitude. The distribution of credible intervals increases with reduced amplitude threshold, and all amplitude thresholds include times with large credible intervals. Note that the phase and amplitude estimates are auto-correlated in time so that each gray dot is not independent of every other dot.

The online version of this article includes the following figure supplement(s) for figure 7:

**Source data 1.** Amplitude and credible interval widths for local field potential (LFP) data.

**Figure supplement 1.** All methods produce similar phase estimates during segments of the theta rhythm with high signal-to-noise ratio (SNR).

The phase estimates and credible intervals for the mu rhythm reveal how this rhythm waxes and wanes in time (example in *Figure 8D and E*; for comparison to phase estimates computed using the alternative methods, see *Figure 8—figure supplements 1 and 2*). When a strong mu rhythm emerges, tight credible intervals surround the mean phase estimate tracking the rhythm of interest; as the mu rhythm wanes, the credible intervals expand. When we restrict analysis to samples with credible interval width less than 25 degrees, the error drops from 40.5 (s.e. 1.1) to 11.2 (s.e. 0.46) degrees

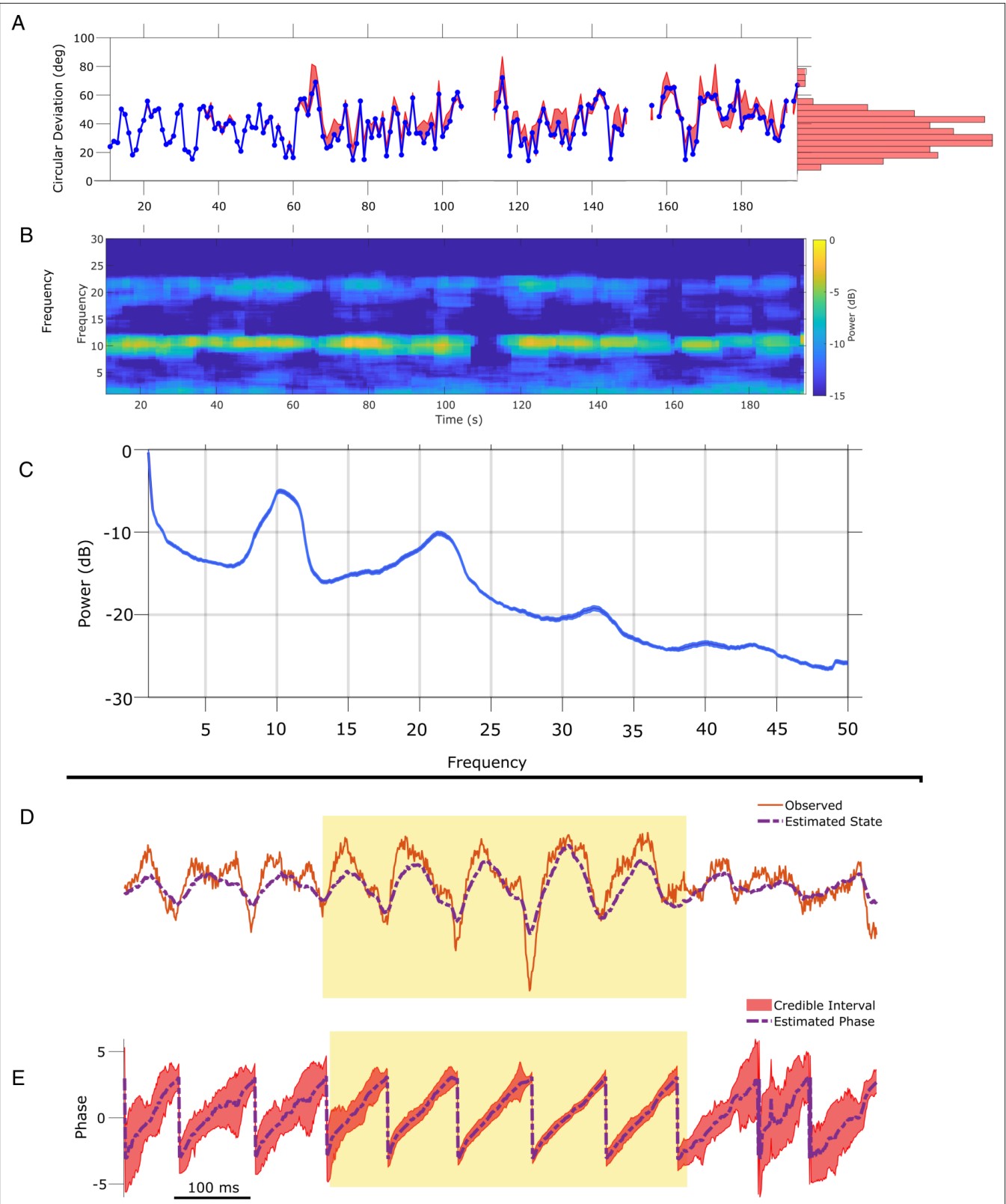

**Figure 8.** State space phase estimate (SSPE) parameters can be estimated from any interval of electroencephalogram (EEG) mu rhythm and credible intervals for mu rhythm phase indicate periods of confidence. (**A**) Phase error for a model fit using data from the first 10 s (blue curve), and the 95 % confidence intervals (red) in the error for phase estimates at time *t* using models fit at times prior to time *t*. Right: histogram of error showing that error remains below 60 degrees in most cases. (**B**) Corresponding spectrogram for the EEG data (multitaper method with window size 10 s, window overlap

*Figure 8 continued on next page*

*Figure 8 continued*

9 s, frequency resolution 1 Hz, and 19 tapers). (**C**) Average power spectral density (PSD) of (**B**) over all times in (**A**). The mean PSD (dark blue) and 95 % confidence intervals (light blue) reveal a peak in the mu rhythm near 11 Hz. (**D**) Example human EEG data (red, solid) with a consistent peak in the mu rhythm (8–13 Hz). The estimated state of the SSPE method (purple, dashed) tracks the observed mu rhythm. (**E**) Phase estimates (purple, dashed) and credible intervals (red, shaded) for the example EEG data in D. When the rhythm appears (yellow shaded region) credible intervals approach the mean phase; when the rhythm drops out the credible intervals expand.

The online version of this article includes the following figure supplement(s) for figure 8:

**Source data 1.** Error and spectrogram for in vivo electroencephalogram (EEG) analysis.

**Figure supplement 1.** All methods produce similar phase estimates during segments of the mu rhythm with high signal-to-noise ratio (SNR).

**Figure supplement 2.** The state space phase estimate (SSPE) method performs optimally across algorithms when tracking the electroencephalogram (EEG) mu rhythm.

**Figure supplement 3.** State space phase estimate (SSPE) captures non-sinusoidality of the mu rhythm.

while still retaining 16.4%  of the data. As with the LFP, we are able to assess certainty in our phase estimates using the credible intervals. Finally, we note that computing the state estimate including both the 10  and 22 Hz components of the model better approximates the non-sinusoidal waveform of the mu rhythm (*Figure 8—figure supplement 3*).

## Pre-stimulus mu rhythm phase predicts MEP amplitude

To illustrate the utility of real-time phase estimation, we again analyze an example EEG recording from *Zrenner et al., 2020*, in which the authors stimulate a subject at random intervals using transcranial magnetic stimulation to elicit a motor evoked potential (MEP) from the right abductor pollicis brevis muscle. We analyze the MEP amplitude to assess the influence of pre-stimulus mu rhythm phase on

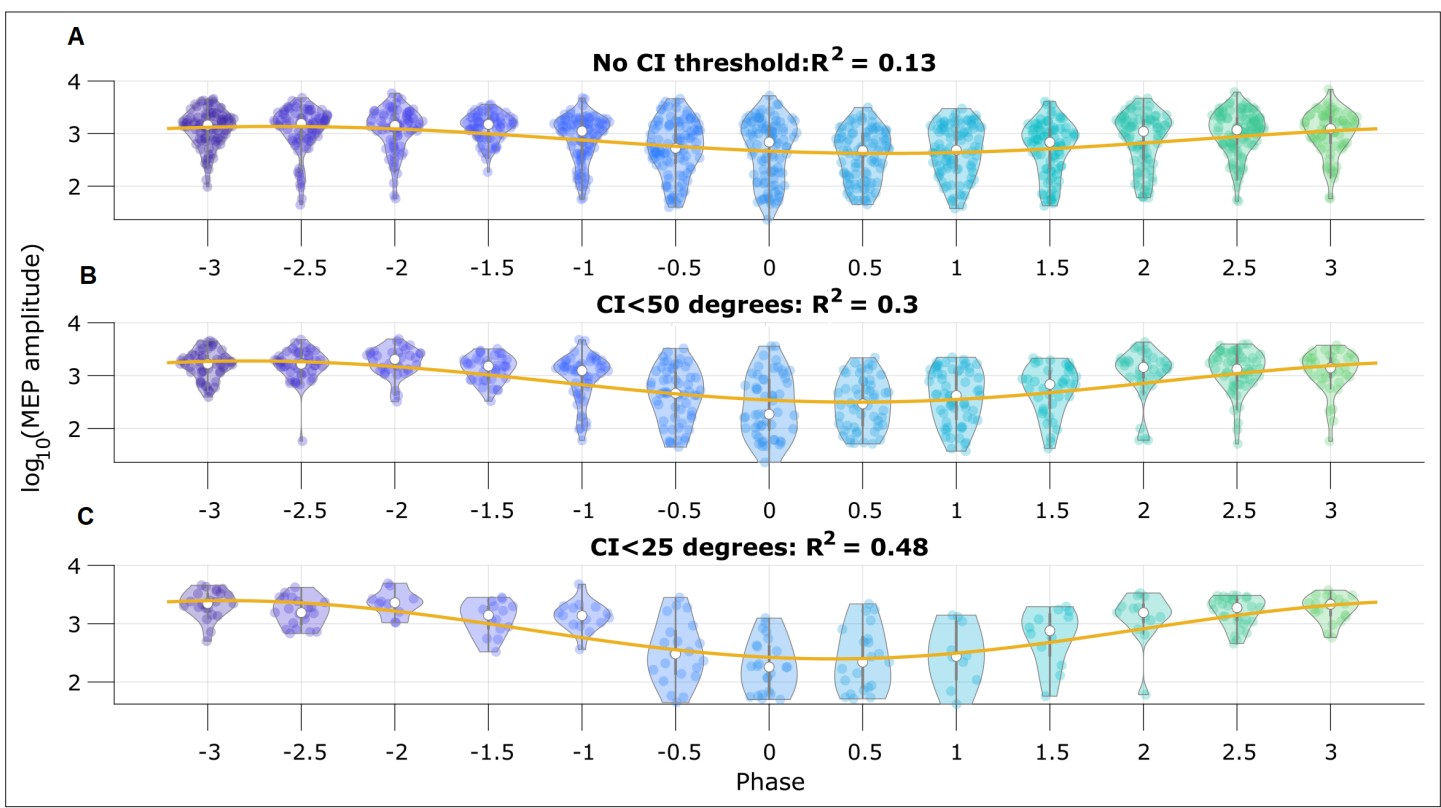

**Figure 9.** Motor evoked potential (MEP) amplitude is predicted by the pre-stimulus state space phase estimate (SSPE) estimated mu rhythm phase. (**A**) Log-transformed MEP amplitude versus phase with no threshold applied to the credible interval width. White dot indicates the median of the MEP amplitude. Yellow line is the best fit circular linear regression line (a phase offset sinusoid) showing that the MEP amplitude is maximum at mu rhythm trough (phase = ±π). (**B,C**) Including only trials with CI width (**B**) less than 50 degrees, or (**C**) less than 25 degrees more clearly demonstrates the MEP amplitude maxima at the mu rhythm trough.

the response (replicating the analysis performed by *Zrenner et al., 2020*). We estimate mu rhythm phase using the SSPE method with model parameters derived from a different subject. We find that the pre-stimulus phase of the mu rhythm, at the instant immediately prior to stimulation, predicts (using circular linear regression) 13% of the log-transformed MEP amplitude (*Figure 9A*). Furthermore, thresholding the trials using credible interval width, we find improved prediction of the MEP amplitude. When using trials with CI width less than 50 degrees, the mu rhythm phase explains 30 % of the variance in the MEP amplitude (*Figure 9B*), and with CI width less than 25 degrees, 48 % of the variance (*Figure 9C*). These results show that modulation of the MEP amplitude depends on the pre-stimulus phase of the mu rhythm, and suggest that accurate estimation of the mu rhythm (e.g., using the SSPE method with narrow CI width) may enhance the evoked motor response.

### Speed and accuracy of TORTE real-time implementation

To assess the validity of the integration of SSPE into TORTE (*Figure 10A*), we generated a 5 Hz sine wave with no additive noise and estimated the phase using the MATLAB and C++ implementations. The parameter estimates for the fitted model were computed on the first 10 s of the data. The computed parameters were nearly identical between the two implementations and the time to estimate the parameters (which, as above, would need to be done only once per experiment) ranged from 2 to 100 s. The phase estimates of the two implementations have a mean circular difference of 2.06 degrees (*Figure 10B*) with a circular standard deviation of 0.65 degree; this variation in phase estimates likely arises from differences in filtering in the TORTE real-time (causal filtering on individual buffers) implementation and the offline MATLAB real-time (acausal filtering across all data) implementation. Further, given that it is a near constant shift in phase, the online learning algorithm in TORTE will correct for the biased phase estimates. We also examined the speed of evaluating different sized buffers with SSPE within TORTE. Across five buffer sizes from 420 to 1700 samples (encompassing 9–37 ms of data, see *Figure 10C*), estimating phase with SSPE took less than 0.35 ms in the worst case. We note that the system latency of TORTE is in the range of a few milliseconds (driven mainly by the USB-based implementation of the current Open Ephys system), whereas the buffers were evaluated on the scale of microseconds. That is, the real-time bottleneck lies in signal amplification and digitization, not this processing. In conclusion, we present a real-time implementation of SSPE that could be easily deployed in various experimental setups.

## Discussion

We have introduced a real-time phase estimator derived from a time series state space model. This approach – the SSPE method – addresses limitations of existing methods (namely, the requirements associated with bandpass filtering in phase estimation) and provides a real-time estimate of confidence in the phase estimate. We demonstrated the benefits of the SSPE method in simulations with varying breadth of the peak rhythm frequency and SNR, a confounding rhythm with a nearby frequency, and phase resets. In two case studies of in vivo data, we showed that re-estimation of model parameters is unnecessary and that the SSPE method provides clear estimates of confidence in phase estimates. We propose that the SSPE serves as a useful tool for real-time phase applications, for example, real-time phase-locked stimulation or real-time phase-amplitude synchrony estimation.

### Narrowband versus broadband signals

While brain rhythms tend to be broadband (*Buzsaki, 2004*; *Roopun et al., 2008*), the most common methods in neuroscience to track a rhythm's phase require appropriate choices of filter parameters (e.g., selection of bandwidth) and generally expect a narrowband oscillation. However, real-time methods that require a narrowband oscillation struggle to capture the phase accurately for broadband rhythms (e.g., *Figure 2*). Moreover, when the broadband rhythm has non-negligible power across all frequencies (in which case the phase cycle is more unstable), the standard non-real-time approach (acausal FIR and Hilbert transform) tracks the phase with limited accuracy. For forecasting needed in real-time analyses, implementing the same assumption (of a broadband peak) in the model may improve phase estimation accuracy. In the SSPE method, the assumption of a narrowband oscillation is not required, as the covariance allows flexibility in modeling peaks of different bandwidths.

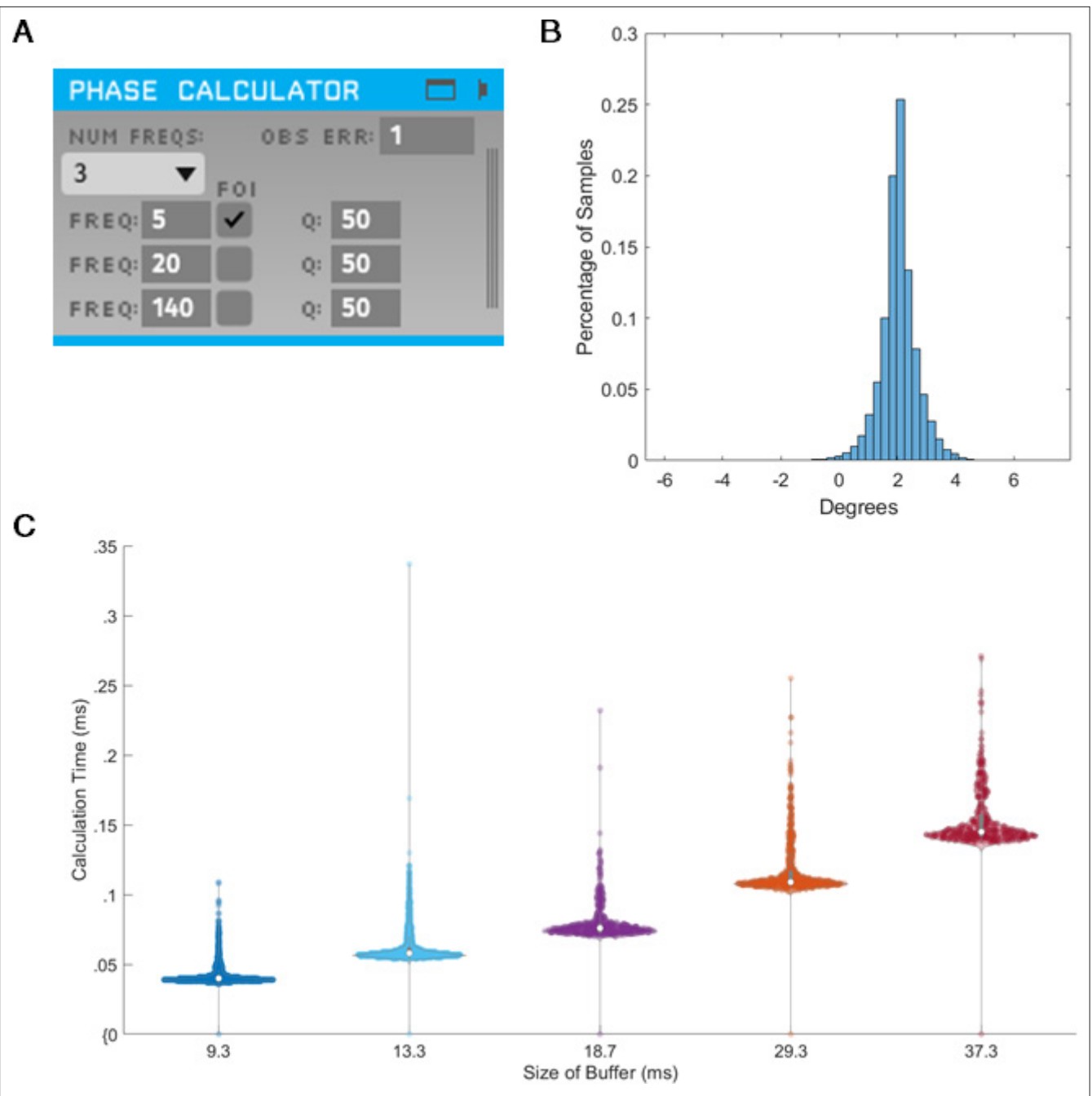

**Figure 10.** State space phase estimate (SSPE) implementation in Toolkit for Oscillatory Real-time Tracking and Estimation (TORTE) is accurate, with negligible latency. (**A**) Open Ephys GUI for using SSPE. The user specifies the number of frequencies to track, the center frequencies to track, the frequencies of interest for phase calculation and output (FOI), variance for the FOI, and the observation error. Observation error determines the effective bandwidth for each frequency. (**B**) Histogram of the circular standard deviation between MATLAB (offline) and TORTE (real-time) implementations of the SSPE. Small variation results from causal low pass filtering in TORTE and acausal filtering in the offline phase estimates. (**C**) Time to evaluate phase versus buffer size. Longer buffer sizes from TORTE require longer calculation time for application of SSPE. However, the calculation time is approximately two orders of magnitude smaller than the buffer size.

## Noise in the signal

By applying a bandpass filter directly on the observed signal, existing real-time phase estimators do not disambiguate the signal and the noise, thus folding the noise into the phase estimate. The impact of structured noise (e.g., a competing rhythm or sudden phase shift) is particularly visible in *Figure 3* and *Figure 4*. To address this, the SSPE approach explicitly separates the signal and noise,

and models two separate components of noise: state noise and observation noise. The SSPE state consists of harmonic oscillators driven by (white) noise. The resulting spectrum consists of broadband peaks at the oscillator frequencies upon a brown aperiodic component, similar to spectra observed in neural field activity. Confounding rhythms (a particular type of noise) can be explicitly modeled and accounted for in the SSPE approach. We consider the observation noise as additive white noise (flat spectrum) due to unknown noise in the measurement process. This model structure is more flexible than methods reliant on bandpass filters, allowing different expectations for the noise (e.g., modeling a confounding rhythm as a pure harmonic oscillator) or adding additional model elements to account for power at higher frequencies (e.g., adding moving average terms; *Matsuda and Komaki, 2017*).

### Detecting a rhythm for phase estimation

To estimate phase for a target rhythm, there should be evidence that the rhythm exists in the data. A few contemporary real-time phase estimators tackle this problem by using an amplitude threshold to decide when a rhythm is present (*Rodriguez Rivero and Ditterich, 2021*; *Rutishauser et al., 2013*). An amplitude criterion depends on accurate knowledge of the distribution of amplitudes possible for the data, relative to the background noise. The SSPE method, in contrast, directly represents the presence/absence of a rhythm in the credible intervals for the phase, as demonstrated for the in vivo data. The result is directly interpretable and permits a more intuitive decision metric (e.g., a threshold depending on the credible interval width) than an arbitrary amplitude criterion. Further, while we expect a central frequency for a rhythm estimated from the data, the Gaussian assumptions of the harmonic oscillator driven by noise allow for the instantaneous frequency to be stochastic. This permits greater flexibility in tracking a rhythm's central frequency compared to prior phase estimators and supports analysis of broadband rhythms with drifting central peaks, and abrupt changes in phase of a fixed rhythm (e.g., *Figure 4*). However, the SSPE method does require identification of target and confounding rhythms for accurate estimation. If details about the rhythms (central frequencies and bandwidth) are unknown, or the target rhythm is known to exist only for short intervals in a long recording, then alternative methods that allow real-time visualization and user-directed intervention in estimation may be pursued (*Rodriguez Rivero and Ditterich, 2021*; *Rutishauser et al., 2013*).

### Computational speed

For real-time phase applications, speed of estimation is critical. We focus here on algorithmic speed as opposed to hardware speed, which is a function of the computing device. Algorithmic speed can be measured at two different points: parameter estimation and application. In the SSPE method and other real-time phase estimators (*Chen et al., 2013*; *Zrenner et al., 2020*), the parameter estimation stage is the slower aspect of phase estimation. Past work has required 10 s of minutes for parameter estimation (*Chen et al., 2013*). Here, for the SSPE method using a window size of 10 s and a sampling rate of 1 kHz, we find that parameter estimation time rarely exceeds a minute or two. In the application stage, we find that speed is orders of magnitude faster than the sampling rate (*Figure 8*), considerably improving on the algorithmic delays present in existing algorithms (*Rodriguez Rivero and Ditterich, 2021*; *Rutishauser et al., 2013*).

### Limitations and future directions

Like all real-time phase estimators, the proposed method possesses specific limitations and opportunities for future improvement. First, the SSPE method treats the generative model for any rhythm as a harmonic oscillator driven by noise. A pure sinusoid is outside this model class, and as such the SSPE method performs slightly worse than filter-based algorithms at tracking the phase of a pure sine. Nonetheless, the error remains small for the SSPE tracking a pure sinusoid in noise (*Figure 2*). We note that a simple alteration in the SSPE model (removing the state covariance $Q$) would allow more accurate tracking of a sinusoid, but not the broadband rhythms more commonly observed in vivo. Second, slow drifts (<0.1 Hz) in the baseline of the data can impair the ability of the SSPE method to track the phase. Slow drifts could be modeled as an additional confounding oscillator, however, estimating its presence would require long segments of time (possibly exceeding 40 s). Alternatively, the SSPE model could be updated to include a model of the slow drift dynamics (perhaps as a piecewise linear term), or a moving average mean trend could be causally removed as we demonstrated with the in vivo EEG analysis. Third, in applying the SSPE method, we assume that the model we estimate

for a rhythm remains appropriate for the extent of the experimental duration. While we showed in two in vivo case studies that this assumption may be reasonable, in other cases this assumption may fail. Rhythms, observed over time, may be better represented by models with changing parameters. Indeed, non-stationarity is an important issue to consider when tracking brain rhythms. The SSPE method is robust (by virtue of the model structure and application of the Kalman filter) to small changes in the central frequency or the bandwidth of a rhythm (e.g., *Figure 2*). However, non-stationary rhythms require new algorithms to be developed (such as the PLSO method in *Song et al., 2020*). An extension of the SSPE method that could potentially track changing central frequencies is to apply an extended Kalman filter (*Schiff, 2012*) that simultaneously estimates the frequencies of interest while filtering the state. Alternatively, the SSPE method could also be extended to implement a switching model that utilizes multiple sets of parameters and switches between parameter sets as necessary, or by refitting the SSPE model as time evolves. Finally, we note that data sampled at a high rate can reduce computational efficiency when fitting parameters for SSPE. To address this, a quasi-Newton optimization – instead of EM, as implemented here – may improve performance, as suggested in *Matsuda and Komaki, 2017*. Determining the optimization framework best suited for practical use remains an important direction for future work.

## Conclusion

We introduced a new algorithm – the SSPE method – to track phase in real-time that addresses several limitations of current methods. We demonstrated in simulation that SSPE performs better than current state-of-the art techniques for real-time phase estimation and comparably, or better, than a commonly used acausal FIR approach. We showed in in vivo data that we can estimate SSPE model parameters and apply them for hundreds of seconds without re-estimation. Finally, we showed how credible intervals from the SSPE method track confidence in the real-time phase estimate, and may provide an intuitive, user-determined decision metric for stimulation at a particular phase. We have provided reference implementations suitable for immediate incorporation into phase-aware in vivo experiments. Future work using the SSPE method in practice may help provide critical evidence on the potential functional importance of phase in neural dynamics.

## Additional information

### Funding

| Funder | Grant reference number | Author |
|---|---|---|
| National Institutes of Health | R01 EB026938 | Anirudh Wodeyar<br>Alik S Widge<br>Uri T Eden<br>Mark A Kramer |
| National Institutes of Health | R01 MH119384 | Alik S Widge |
| National Institutes of Health | R01 MH123634 | Alik S Widge |

The funders had no role in study design, data collection and interpretation, or the decision to submit the work for publication.

### Author contributions

Anirudh Wodeyar, Conceptualization, Formal analysis, Investigation, Methodology, Software, Visualization, Writing - original draft, Writing – review and editing; Mark Schatza, Data curation, Formal analysis, Methodology, Software, Visualization, Writing – review and editing; Alik S Widge, Conceptualization, Funding acquisition, Investigation, Methodology, Supervision, Writing – review and editing; Uri T Eden, Conceptualization, Funding acquisition, Methodology, Supervision, Writing – review and editing; Mark A Kramer, Conceptualization, Funding acquisition, Investigation, Methodology, Resources, Supervision, Writing – review and editing

### Author ORCIDs
Anirudh Wodeyar (iD) http://orcid.org/0000-0003-2577-5139
Alik S Widge (iD) http://orcid.org/0000-0001-8510-341X

### Decision letter and Author response
Decision letter https://doi.org/10.7554/eLife.68803.sa1
Author response https://doi.org/10.7554/eLife.68803.sa2

---

## Additional files

### Supplementary files
• Transparent reporting form

### Data availability
All data generated or analyzed during this study, or were used to create the figures are included in the supporting files, or are available through already public data archives (https://gin.g-node.org/bnplab/phastimate, https://doi.org/10.6084/m9.figshare.14374355).

The following previously published datasets were used:

| Author(s) | Year | Dataset title | Dataset URL | Database and Identifier |
|---|---|---|---|---|
| Zrenner C, Galevska D, Nieminen JO, Baur D, Stefanou MI, Ziemann U | 2020 | Phastimate | https://gin.g-node.org/bnplab/phastimate | Phastimate, 116761 |
| Wodeyar A, Schatza M, Widge A, Eden UT, Kramer MA | 2021 | SSPE data | https://figshare.com/articles/dataset/SSPE_data/14374355 | figshare, 10.6084/m9.figshare.14374355 |

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
