## [Decision Letter]

**Acceptance summary:**

This paper will be of interest to theoretical and experimental neuroscientists. It proposes a novel approach based on state space modeling to estimate the phase of neural signals like EEG and LFP in real-time. It is expected to help drive the improvement and proliferation of phase-related concepts in neurobiology.

**Decision letter after peer review:**

Thank you for submitting your article "A State Space Modeling Approach to Real-Time Phase Estimation" for consideration by *eLife*. Your article has been reviewed by 3 peer reviewers, and the evaluation has been overseen by a Reviewing Editor and Laura Colgin as the Senior Editor. The following individuals involved in review of your submission have agreed to reveal their identity: Milad Lankarany (Reviewer #1).

Essential revisions:

1) Consider the impact of noise in the phase estimation, include specifications of the Kalman filter and its robustness, and consider the performance of the estimated phase relative to other methods.

2) Demonstrate the advantage of using this method for exploring the link between rhythms and behaviour and for phase-locked stimulation, and separate applications that require real time phase estimation from those that can be performed offline using acausal methods.

3) Provide some help to readers from non-technical background by linking the method back to applications and in-vivo data more.

4) Expand details regarding computing credible intervals, slow drift considerations, and physiological motivation of the simulations.

Further details are provided below in the 'Recommendations for the authors'.

*Reviewer #1 (Recommendations for the authors):*

I believe the proposed method in this paper is very novel and timely for the neuroscience community. The paper is well organized and written very clearly. I have some comments that might improve the quality of this paper.

1. Impact of Noise in the phase estimation. Although there is a section in the Discussion related to Noise in Signal, the authors did not address the impact of Signal to Noise Ratio (SNR) on the performance of their algorithms. I think this will add value to your algorithm, specifically because the Kalman Filter can be much more reliable than other methods in this context.

2. Specifications of the Kalman Filter and its robustness. specifications of the Kalman filter (KF) should be described in the Method Section. For example, the choice of state and observation covariance matrices (Q and R) should be clarified. In this regard, the authors can discuss the robustness of the KF in phase estimation.

3. For in-vivo data, how the performance of the estimated phase might be compared to other methods (used in the simulation study)? It can be helpful to visually compare the estimated phase calculated by SSPE and other methods which were used in the simulation study.

*Reviewer #2 (Recommendations for the authors):*

I think the manuscript can be improved by

1) clearly demonstrating the advantage of using this method for exploring the link between rhythms and behaviour and for phase-locked stimulation. The authors should separate applications that require real time phase estimation from those that can be performed offline using acausal methods.

2) Building on my previous point:

– In figure 2, proposed method clearly outperforms others for signals generated using the "state space model". How representative is this for in-vivo recordings?

– In figure 4, it is demonstrated that other methods cannot track phase-reset, however, is this offset sustained or is it a transient slip in phase estimation for one cycle. In other words, if we zoom out in time, do the other methods converge back to true phase? And if so, when would it be critical to track phase-reset for one cycle?

– The performance of SSPE should be demonstrated with respect to other methods for in-vivo data (figures 5 and 7) so that the reader can evaluate the utility of this method.

3) The paper is well written but comes across as very technical. Readers from non-technical backgrounds may struggle with following some of the concepts. This could be improved by linking the method back to applications and in-vivo data more, and also by reviewing the language used in the methods segment.

*Reviewer #3 (Recommendations for the authors):*

1) Credible intervals

An interesting aspect of the manuscript is the possibility of computing credible intervals, making it possible to establish bounds on time points when the phase can be estimated with sufficient accuracy. In previous approaches, this is usually done via an amplitude threshold, with the assumption that a high amplitude will also result in high phase estimation accuracy. In Figure 6 the authors therefore study the relationship between amplitude and credible intervals. It is tough to interpret this evidence; the argument is that for an amplitude threshold using the 65th percentile the range of credible intervals is from the minimum to maximum possible. But looking at the major bulk of probability mass, it's centered around intervals with an acceptable looking range? Could more details regarding computation of confidence intervals be provided, e.g. are they symmetric? Figure 6C is a scatter plot with lots of points that are correlated, maybe it would be good to visualize this in a different way reducing the number of points per cycle? Can the credible intervals be related to error in degrees? They look pretty negligible in the left part of the figure (given that the real-time procedure also introduces a lag, Figure 8B),is the plotted x-range the relevant one? How does SNR/amplitude relate to phase accuracy to credible intervals? If the spread of credible intervals can be largely explained by amplitude (aka high correlation between credible interval width and amplitude), then maybe this is not such a big factor?

2) Slow drifts

A practical reason why existent algorithms rely on buffering is the frequent existence of slow drifts in the data, for which detrending needs to be applied. For the EEG data, the following methodological detail of the current approach is given "Before real-time phase estimation, we high-pass filter the data at 0.5 Hz to remove slow drifts." This procedure is not possible when estimating the phase in real-time. The authors claim the benefit of their approach is the absence of needing buffers, but in the light of this I am not sure about whether this is the case. In the discussion, the authors discuss the possibility of modeling the slow drifts via an additional rhythm in the state space, but state that long segments of data are needed for this. In my view, this is a crucial aspect of any algorithm targeting real-time phase and more information and analysis resolving slow drifts would greatly improve the actual applicability of the present approach.

3) Data generation process and simulations

I find some of the simulations not sufficiently motivated from a physiological perspective.

– Two simultaneous rhythms with nearby frequencies: In this scenario the authors simulate the signal as a summation of two sinusoids with a frequency offset with a fixed phase shift. This will result in a signal with amplitude modulation with the envelope given by the difference of the frequencies, resulting in the appearance of oscillatory bursts. I would argue that for physiological signals that present like this, it is not of interest to track a single frequency in this signal, as e.g. the trough phase of this compound signal corresponds to a specific physiological state. I would also suggest showing the amplitude for this case in Figure 3. Possibly the simulation could be modified to be better physiologically motivated, as they are certainly cases where there are neighboring rhythms which present on the same electrode through volume conduction, but it is unclear (but would be of interest) to me if the proposed method could be helpful in this case.

– From my perspective phase is only well-defined if there are clear-cut oscillatory bursts. In that sense, the signal given as an example in Figure 2A iv does not have clear-cut oscillatory states, with no clear spectral peak and the true phase (blue curve) changing rapidly. The authors show that the algorithm performs well here as this is exactly according to their data-generating process assumptions (e.g. Figure 2C right-most), whereas for the other algorithms errors accumulate especially in fast phase changing periods. In my view, this simulated situation conceptually does not capture processes that are relevant to investigating neural oscillations in the brain, which would follow a conceptual model more similar to: multiplicative 1/f contribution * oscillator + additive 1/f contribution, so that presumably corresponds to the presented filtered pink noise case here. Can the authors clarify the motivation for this simulated case?

– The authors speak of broadband oscillations, for instance relating to the mu-rhythm (l. 554). In my view this is a narrowband oscillation with a clear peak judging from the power spectrum, with the broadness of the peak influenced by the presence of strong amplitude envelope modulations. Can authors clarify their criteria for defining an oscillation as broadband, in my view this is non-standard terminology.

– One limitation of band-pass filtered approaches that the authors mention is non-sinusoidality (l. 48). In my view, this is manifested in differences in waveform shape, e.g. the arc-shape of the mu-rhythm in Figure 7D. It does not seem that problems in phase estimation for this type of signals (e.g. non-uniform phase-velocity) is addressed by the method as the estimated state does not capture the arc-shape of the rhythm. Can the authors clarify in what way they contribute to resolve this limitation?

4) I would welcome additional details in the text regarding following methodological procedures, which should yield a more self-contained manuscript:

– While bandpass filtering and Hilbert-transform are quite common operations for electrophysiologists, a state space approach may be unfamiliar to the typical reader. Possibly it would be helpful to have clearer paragraphs describing the framework (aka rework l. 98ff), I think the attempt here was to first explain the procedure without formulas, which appear further down, I am not quite sure that this succeeded in providing clarity. Maybe it would be helpful to first make an example with only 1 rhythm, so that dimensionality of the involved entities is not as complex as for tracking N rhythms. Maybe it would also be helpful to extend Figure 1 with a conceptual diagram. How is the number and frequency of oscillators chosen for empirical data?

– E.g. regarding the parameter estimation the reader is referred to the supplementary material of Soulat et al. 2019. Can an intuition about this procedure be given? I find this interesting in the context that a quite large range for expected computation time is given (2-100 s, l. 573). Can the authors comment on factors influencing this? Maybe a more detailed description of the procedure will be helpful for understanding this.

– Currently descriptions of the comparison algorithms are lacking, e.g. what is the general procedure and difference between Zrenner and Blackwood algorithm, what are the crucial parameters here? The authors give code-accessibility as the main reason for selecting those two specific instances. Because recently a lot of different methods (mostly variants involving band-pass filtering) were proposed for estimating the phase in real-time, maybe those 2 selected ones could be better discussed in the context of others in the literature.

– typo, l.55 extant -> existant

– the dotted lines for phase estimates may be not optimal visually (e.g. Figure 6B), maybe shading of credible intervals would look nicer?

– Can the authors provide more details on how the power spectra were estimated? The simulation ones look like Welch's method and the data ones like multitaper were used.

– If I understood correctly, the employed two-sided t-test is calculated using lots of correlated data points (using all time points in an interval), resulting in reduced degree of freedom. I think the evidence that the SSPE approach works better here is actually strong, but the t-test looks slightly weird because of wrong degrees of freedom, distracting from the result. Maybe the information from table 1 can just be plotted in a distributional way, more like Figure 2C?

[Editors' note: further revisions were suggested prior to acceptance, as described below.]

Thank you for resubmitting your work entitled "A State Space Modeling Approach to Real-Time Phase Estimation" for further consideration by *eLife*. Your revised article has been evaluated by Laura Colgin (Senior Editor) and a Reviewing Editor.

The manuscript has been considerably improved and this interesting paper provides inspiration for the field, bringing state space models and phase estimation for neural recordings together. However, there are a few remaining points that are mainly related to presentation results that could be improved. It will not be sent back to the reviewers for final evaluation.

*Reviewer #3 (Recommendations for the authors):*

I thank the authors for extensive replies to my comments, which have clarified many aspects and made the method description more accessible. I like the new waveform shape figure showing the simultaneous tracking of the base + harmonic frequency. I find the SSPE method interesting from a principal perspective, but am not yet sure in what way the performance of the algorithm is superior to other algorithms in physiologically relevant cases. My remaining comments:

1) Plausibility of simulations R3.Q4

– The state space model simulations (Figure 2A.iv) generating model are not a common data generation model for the rhythms the authors actually investigate (theta and mu). The newly added citations are for the γ-rhythm (and a quite obscure EEG paper). -> I would say this is not a generally accepted data generation model for mu and theta. The fact that the SSPE performs well on data generated from the exact state space model does not seem to entirely relevant for the chosen application, with the simulation lacking stability in cycle periods that is present for mu and theta. Possibly, a note can be added in the Discussion section in the narrow-band vs broad-band signal section, reflecting on broad-band phase would be beneficial.

– The filtered pink noise seems to result in a broader spectral peak (green line in Figure 2B.ii) than is present in the empirical data. As the band-pass filter parameters of the alternative algorithms are possibly adjusted for more narrowband signals, not adjusting the bandwidth before e.g., doing a Hilbert transform may yield an unfair comparison. Can you comment on that?

– My suggested approach using multiplicative 1/f corresponds to the second of the simulations provided by the authors in R3.A4. To clarify this, I attach some python lines at the end to show a simulated signal, if there is interest (parameters to be adjusted). The phase estimation errors in the provided plot in the response seem really low, so maybe the oscillatory SNR is too high for this simulated case (also the amplitude modulation is not really prominent?) I find it confusing that the SSPE method does not yield any benefit here. I am not requesting specifically for this type simulation to be included, but I would like to understand generally the conditions under which SSPE yields better results than the alternative algorithms.

From my understanding, SSPE should yield benefits for phase estimation in intermediate SNR ranges, but at the moment the authors have added more evidence that all algorithms show very similar behavior at high SNR (Figure Supplement I for Figure 7 and 8) and the simulation type provided in Figure 5 has some issues with phase stability as it is filtered pink noise (see comments above) and the modelled SNR range (there is not much change in phase estimation errors which I would have expected to vary with increasing SNR).

R3.Q3:

I can acknowledge the scenario of the two simultaneous rhythms with nearby frequencies which the authors attribute to two different rhythms instead of seeing as amplitude modulated bursts. For this case, maybe the modeling scenario would be more general, if the rhythms had varying phase lag (random for each trial) instead of a fixed phase lag (pi/4), which seems like the case in which the rhythms are most easily distinguishable.

2) The relationship between credible intervals and amplitude measures. R3.Q1.

– The authors argue for a dissociation between credible intervals and amplitude values, with evidence provided by e.g., Figure 7. Given the nature of the data, with oscillations occurring in bursts and not as a continuous oscillation, I am wary of phase estimates for time points where there is no oscillation (e.g., for the theta data, oscillation probability is heavily dependent on animal speed and there will be periods in the data without an oscillation). My main point here is that credible intervals seem to have a correspondence to amplitudes in a range where an oscillation is clearly present (high amplitude) and phase is not defined when an oscillation is absent. Sure, the acausal FIR will return a value for every time point, but it's not necessarily meaningful in a physiological sense. Taking this as the ground truth is problematic for the credible interval and amplitude measures dissociation. For instance, is there a discernible oscillation for the data in Figure 7 for an amplitude criterion > 65th percentile? Is a wide or narrow credible interval around a value that is not defined meaningful?

Figure 7C is a scatter plot with lots of points that are correlated, maybe it would be good to visualize this in a different way reducing the number of points per cycle

possibly by subsampling or a 2D histogram. In the current presentation, the plot basically just shows the range of the data, amplifying low-occurrence data points.

This suggests that, at low SNR, an arbitrary amplitude threshold (e.g., amplitude > 1) would eliminate all data at SNR=1, and preserve all data at SNR=2.5.

Typically (e.g., in the Zrenner studies), amplitude is defined as a relative criterion, e.g., with percentile threshold or mean amplitude +- 2.5 SD using some training interval, so I am not sure about this argument.

– We find that fixed amplitude thresholds produce wide ranges of credible intervals (Figure 7C.ii). This is especially apparent at lower amplitude thresholds; choosing a threshold of 65% results in credible intervals that range from 0.042.29 (minimum) to 174.75 (maximum) degrees.

Can authors list the 95% range here, rather than min/max?

– Figure 5: Can a spectrogram can also be plotted in this figure?

I would expect the height of the spectral peak over the 1/f-contribution to vary here.

3) Code availability

I wanted to check the used bandpass filter parameters (since it's not described for the Blackwood algorithm) and noticed that the linked repository does not feature the code to recreate the plots in the manuscript, possibly it would be helpful for other researchers to provide that, also for comparing to other algorithms in the future.

---

## [Author Response]

Essential revisions:1) Consider the impact of noise in the phase estimation, include specifications of the Kalman filter and its robustness, and consider the performance of the estimated phase relative to other methods.

We address these suggestions in detail below.

2) Demonstrate the advantage of using this method for exploring the link between rhythms and behaviour and for phase-locked stimulation, and separate applications that require real time phase estimation from those that can be performed offline using acausal methods.

We address these suggestions in our response in R2.A2.

3) Provide some help to readers from non-technical background by linking the method back to applications and in-vivo data more.

Please see our responses in R3.A7 for a detailed response to this suggestion.

4) Expand details regarding computing credible intervals, slow drift considerations, and physiological motivation of the simulations.

Please see our responses in R2.A3, R3.A1, R3.A2, R3.A3 and R3.A4 where we address these points.

Further details are provided below in the 'Recommendations for the authors'.Reviewer #1 (Recommendations for the authors):I believe the proposed method in this paper is very novel and timely for the neuroscience community. The paper is well organized and written very clearly.

Thank you for your kind comments, they helped us further describe how performance links to signal-to-noise ratio and show the performance of SSPE relative to other methods.

I have some comments that might improve the quality of this paper.1. Impact of Noise in the phase estimation. Although there is a section in the Discussion related to Noise in Signal, the authors did not address the impact of Signal to Noise Ratio (SNR) on the performance of their algorithms. I think this will add value to your algorithm, specifically because the Kalman Filter can be much more reliable than other methods in this context.

R1.A1. As recommended by the Reviewer, we have updated the manuscript to include a new simulation study characterizing the impact of signal to noise on real time phase estimation. We show that the SSPE method outperforms the other real time methods, and that the credible intervals (CI) vary although signal amplitude is fixed (in response to R3.Q1). We have updated the Results and Methods as follows:

Results, after Section Phase estimation following phase reset

“Phase estimation with different signal-to-noise ratios: To demonstrate the impact of noise on phase estimation we simulate signals with different signal-to-noise ratios. To generate a signal we filter pink noise into the band of interest (FIR filter, 4-8 Hz, “Filtered Pink Noise”, see Methods), and then add additional (unfiltered) pink noise. The ratio of the standard deviations of the signal and noise defines the signal-to-noise ratio; we modulate the standard deviation of the signal to achieve the desired SNR. Finally, we normalize the full observation so that amplitudes across simulations are consistent. In these simulations the SNR varies from 1 (signal and noise have the same total power) to 10 (signal has 100 times the total power of the noise); see example traces of the observation under each SNR in Figure 5A.i – iv. We repeat each simulation 1000 times with different noise instantiations for each signal to noise ratio.

We compute two metrics to characterize the impact of signal-to-noise on phase estimation. First, we consider the credible intervals estimated by the SSPE, here measured as the average credible interval width throughout a simulation. As expected, the credible intervals track with the SNR (Figure 5.B.ii); signals with higher SNR correspond to smaller credible intervals. Computing the average amplitude (as derived from the acausal FIR, see Methods: “Non-Causal Phase Estimation”) throughout each simulation, we find no overlap between the distributions of amplitudes at SNR=1 and SNR=2.5 (5B.ii), while the distributions of credible intervals overlap. This suggests that, at low SNR, an arbitrary amplitude threshold (e.g., amplitude > 1) would eliminate all data at SNR=1, and preserve all data at SNR=2.5. Alternatively, using the credible interval width allows identification of transient periods of confidence (e.g., credible interval width < 40 degrees) at both SNR values. We also note that the CI width – in units of degrees – provides a more intuitive threshold than the amplitude, whose intrinsic meaning at any particular value is only understood relatively. These results suggest that, in this simulation study, amplitude serves as a less accurate criterion for confidence in phase compared to the CI width.

Second, we consider the error in phase estimates using four different phase estimation methods (Figure 5C). We find that the error for the SSPE method reduces more quickly than for the other methods, and at a SNR of 1, the SSPE method is comparable to or better than the contemporary methods of real-time phase estimation. At SNR = 10, the error in the SSPE phase estimation is comparable to the acausal FIR, while the alternative real-time algorithms continue to perform with error similar to an SNR of 1. These results suggest that, for the two existing real-time phase estimation methods, broadband rhythms are difficult to track even with high SNR, consistent with Figure 2C.”

Methods, Simulated data, following the sub-section Phase Reset;

Signal to Noise Ratio

“To investigate the impact of signal-to-noise ratio (SNR) on phase estimation, we again simulate a filtered pink noise signal (centered at 6 Hz) with added pink noise (the third type of rhythm simulated in Methods: Narrow Band to Broad Band Rhythms). To modulate the SNR, we first compute,SNR0 = σsignal / σnoise

where σsignal (σnoise) is the standard deviation of the filtered pink noise signal (added pink noise). We then divide the filtered pink noise signal by SNR0, multiply the filtered pink noise signal by the chosen SNR (1, 2.5, 5, 7.5, 10), and add this scaled signal to the pink noise.”

2. Specifications of the Kalman Filter and its robustness. specifications of the Kalman filter (KF) should be described in the Method Section. For example, the choice of state and observation covariance matrices (Q and R) should be clarified. In this regard, the authors can discuss the robustness of the KF in phase estimation.

R1.A2. As recommended, we have updated the Methods to clarify the choice of the state and observation covariance matrices (Q and R), as well as the starting values for the state and state covariance as follow:

Methods, following equation (5):

“We note that the state (Q) and observation (R) covariance matrices are estimated from the data, as described below.”

Methods, Section Real-time Phase Estimation:

”We now define the stages of Kalman filtering. The state is initialized with zeros and the state covariance to a diagonal matrix with 0.001 along the diagonal. To predict the future state and state estimation error …”

We also now discuss the robustness of the Kalman filter in phase estimation:

Methods, Section Real-time Phase Estimation:

“ … With these model parameters estimated and fixed, we then apply a Kalman filter to predict and update the state estimates, and estimate the phase and amplitude for each oscillator, each representing a different rhythm, for every sample (Figure 1). We note that the Kalman filter is an optimal filter, i.e., in the presence of Gaussian noise, the Kalman filter achieves minimum error and is an unbiased estimate of the mean phase. However, if the noise is not Gaussian, the Kalman filter remains the optimal linear filter for phase estimation under this model. Given the stochastic frequency modulation that is possible under the model, the Kalman filter is robust to small shifts in the central frequency of the rhythm.”

3. For in-vivo data, how the performance of the estimated phase might be compared to other methods (used in the simulation study)? It can be helpful to visually compare the estimated phase calculated by SSPE and other methods which were used in the simulation study.

R1.A3. To address this, we have updated the Results to include the estimated phase computed with the other real-time phase estimation methods. We show that phase estimates converge when the rhythm has a high signal-to-noise ratio (SNR), but diverge at low SNR for both the theta and the mu rhythm. Additionally, we show error estimates for both the LFP theta and the EEG mu rhythm data, demonstrating that the SSPE performs comparably to the acausal FIR on the data. We note that the error estimate critically depends on the generative model assumed for the in vivo signal (the SSPE method and existing methods assume different underlying models for the rhythm). We have updated the Results in the revised manuscript as follows,

Results, Section Example in vivo application: rodent LFP:

…”However, comparing the estimated phase to the true phase (see Methods), we find that the phase error tends to remain consistent in time (blue curve in Figure 6A; circular standard deviation 95 percent interval = [33.11, 75.39]; for error of phase estimates computed using alternative methods, see Figure Supplement 6-I). “…

“To illustrate this, we show the phase and credible interval estimates of the SSPE method for an example 3 s interval of the LFP data (Figure 7; for comparison to phase estimates computed using the existing methods, see Figure Supplement 7-I). We note that, …”

Reviewer #2 (Recommendations for the authors):I think the manuscript can be improved by1) clearly demonstrating the advantage of using this method for exploring the link between rhythms and behaviour and for phase-locked stimulation. The authors should separate applications that require real time phase estimation from those that can be performed offline using acausal methods.

R2.A2. To address this question, we again utilize the dataset shared by Zrenner and colleagues in Zrenner et al. 2020. There, the authors stimulate using transcranial magnetic stimulation (TMS) at random phases of the mu rhythm and use a post-hoc causal analysis to examine whether there is modulation of the motor evoked potentials’ (MEP) amplitudes as a function of the phase of the mu rhythm. They show that using their real-time phase estimation method, MEP amplitude is selectively higher closer to the trough of the mu rhythm. We test whether the SSPE method can replicate these findings. To do so, we apply the SSPE method, using parameters estimated on a different subject, to the pre-stimulus EEG data to estimate the phase at the moment of TMS stimulation. We find that phase as estimated from the SSPE method is a strong predictor of the MEP amplitude (when fitting a circular linear regression model), with the peak close to the mu rhythm trough as predicted.

We add a new figure and results to the Results section of the manuscript, as follows:

Results: Pre-stimulus Mu-Rhythm Phase Predicts MEP amplitude

“To illustrate the utility of real-time phase estimation, we again analyze an example EEG recording from (Zrenner et al., 2020) in which the authors stimulate at random intervals using transcranial magnetic stimulation (TMS) a subject using to elicit a motor evoked potential (MEP) from the right abductor pollicis brevis muscle. We analyze the MEP amplitude to assess the influence of pre-stimulus mu rhythm phase on the response (replicating the analysis performed by Zrenner et al. 2020). We estimate mu rhythm phase using the SSPE method with model parameters derived from a different subject. We find that the pre-stimulus phase of the mu rhythm, at the instant immediately prior to stimulation, predicts (using circular linear regression) 13 percent of the log-transformed MEP amplitude (Figure 9A). Furthermore, thresholding the trials using credible interval width, we find improved prediction of the MEP amplitude. When using trials with credible interval width less than 50 degrees, the mu rhythm phase explains 30% of the variance in the MEP amplitude (Figure 9B), and with credible interval width less than 25 degrees, 48% of the variance in the MEP amplitude (Figure 9C). These results show that modulation of the MEP amplitude depends on the pre-stimulus phase of the mu rhythm, and suggest that accurate estimation of the mu rhythm (e.g., using the SSPE method with narrow credible interval width) may enhance the evoked motor response.”

Further, we have ensured that in the manuscript, we clarify whether real-time analysis is required:

Phase Estimation Following Phase Reset: “… Further, accurate tracking of phase resets would provide better evidence for the relevance of phase resetting in generating event related potentials (Sauseng et al., 2007) – an analysis that could also be performed acausally.”

2) Building on my previous point:– In figure 2, proposed method clearly outperforms others for signals generated using the "state space model". How representative is this for in-vivo recordings?

R2.A3. To clarify this point, we have updated the manuscript to motivate the state space model as representative of features in in vivo recordings as follows:

Methods: Narrow Band to Broadband Rhythms: “For the final rhythm we simulate a signal yt from the SSPE model (see Methods: State Space Model Framework) which is non-sinusoidal and broadband. … The state in the state space model is an auto-regressive model of order 2 (West 1997a) (equivalently, a damped harmonic oscillator driven by noise), which has been proposed to capture features of γ rhythms in visual cortex and rhythms in EEG (Franaszczuk and Blinowska 1985, Xing et al., 2012, Burns et al. 2010, Spyropoulos et al. 2020). The addition of observation noise under the state space model can be analogized to measurement noise. Thus, the state space model provides a simple possible model for brain rhythms observed in electrophysiological data.”…

New references:

West, M. (1997a) Time series decomposition. Biometrika, 84, 489-494

Xing, D., Shen, Y., Burns, S., Yeh, C. I., Shapley, R., and Li, W. (2012). Stochastic generation of γ-band activity in primary visual cortex of awake and anesthetized monkeys. Journal of Neuroscience, 32(40), 13873-13880a.

Burns, S. P., Xing, D., Shelley, M. J., and Shapley, R. M. (2010). Searching for autocoherence in the cortical network with a time-frequency analysis of the local field potential. Journal of Neuroscience, 30(11), 4033-4047.

Franaszczuk, P. J., and Blinowska, K. J. (1985). Linear model of brain electrical activity—EEG as a superposition of damped oscillatory modes. Biological cybernetics, 53(1), 19-25.

Spyropoulos, G., Dowdall, J. R., Schölvinck, M. L., Bosman, C. A., Lima, B., Peter, A., … and Fries, P. (2020). Spontaneous variability in γ dynamics described by a linear harmonic oscillator driven by noise. bioRxiv, 793729.

– In figure 4, it is demonstrated that other methods cannot track phase-reset, however, is this offset sustained or is it a transient slip in phase estimation for one cycle. In other words, if we zoom out in time, do the other methods converge back to true phase? And if so, when would it be critical to track phase-reset for one cycle?

R2.A4. Indeed, it is the case that the methods all return to accurately track the rhythm after some time. This simulation suggests that, when stimuli appear for a human or non-human animal during normal behavior (when phase resets might occur), the SSPE method continues to perform well. In fact, recent work attempting to perform phase stimulation based on the theta rhythm has been forced to avoid segments with phase resets to reduce error in real-time phase estimation (Gordon et al. 2021). While bandpass filter approaches struggle with phase estimation in the presence of phase resets as a function of the window size used, the SSPE method (by virtue of the Kalman Filter) continues to perform optimally.

To address this important question, we include additional analysis and explanation in the revised manuscript as follows:

Results: Phase estimation following phase reset: “…Phase resets are common and well documented in electrophysiological recordings (Fiebelkorn et al., 2011; Makeig et al., 2004). Recent work delivering electrical stimulation based on phase of the theta rhythm has been forced to avoid segments with phase resets to reduce error in real-time phase estimation (Gordon et al., 2021). Further, accurate tracking of phase resets …

…“These results demonstrate that the existing causal and acausal methods implemented here are unable to estimate the phase accurately immediately after a phase reset. To determine how quickly each method returned to accurate phase estimation, we first estimated the pre-stimulus average error for each method, prior to the first phase reset, over a 500 ms interval. We then determined the time between each subsequent phase reset and the moment when the error reduced to 1.5 times the pre-stimulus error. Note that the level of pre-stimulus error differs across methods, so, for example, we determine when the SSPE converges to a lower error than the Blackwood method (see top row of Table 1). We find that each method eventually converged to accurate phase estimation after a phase reset, with the SSPE method converging most rapidly (Table 1). The differences between methods likely result from the windowing used by each method when estimating the phase; smaller windows will lead to less impact of the phase reset on the error (as seen with the Blackwood approach converging more quickly than the Zrenner method), with the SSPE needing a window of a single sample.”

– The performance of SSPE should be demonstrated with respect to other methods for in-vivo data (figures 5 and 7) so that the reader can evaluate the utility of this method.

R2.A5. As recommended, we now include this comparison in the revised manuscript. Please see R1.A3 for a detailed response (including new figures) to this comment.

3) The paper is well written but comes across as very technical. Readers from non-technical backgrounds may struggle with following some of the concepts. This could be improved by linking the method back to applications and in-vivo data more, and also by reviewing the language used in the methods segment.

R2.A6. To address this comment, we have updated the manuscript to include additional intuitive descriptions (including a new schematic diagram) in the Methods; please see R3.A7 for details. In addition, we now include a new in vivo application; please see R2.A2 for details.

Reviewer #3 (Recommendations for the authors):1) Credible intervalsAn interesting aspect of the manuscript is the possibility of computing credible intervals, making it possible to establish bounds on time points when the phase can be estimated with sufficient accuracy. In previous approaches, this is usually done via an amplitude threshold, with the assumption that a high amplitude will also result in high phase estimation accuracy. In Figure 6 the authors therefore study the relationship between amplitude and credible intervals. It is tough to interpret this evidence; the argument is that for an amplitude threshold using the 65th percentile the range of credible intervals is from the minimum to maximum possible. But looking at the major bulk of probability mass, it's centered around intervals with an acceptable looking range? Could more details regarding computation of confidence intervals be provided, e.g. are they symmetric? Figure 6C is a scatter plot with lots of points that are correlated, maybe it would be good to visualize this in a different way reducing the number of points per cycle? Can the credible intervals be related to error in degrees? They look pretty negligible in the left part of the figure (given that the real-time procedure also introduces a lag, Figure 8B),is the plotted x-range the relevant one? How does SNR/amplitude relate to phase accuracy to credible intervals? If the spread of credible intervals can be largely explained by amplitude (aka high correlation between credible interval width and amplitude), then maybe this is not such a big factor?

R3.A1. We address these important points in the updated manuscript as follows:

Updates to original Figure 6 (now Figure 7)

To help improve the interpretation of original Figure 6 (now Figure 7), we have updated the x-axis to show values in degrees. Additionally, we have edited the Figure caption to reflect the autocorrelation of points across time:

Figure 7 … (C.i): Phase credible intervals versus theta rhythm amplitude for each time point (gray dot). On the x-axis we plot an example cycle of a rhythm with credible intervals, see x-axis of (C.ii) for numerical values of credible intervals. Note that the phase and amplitude estimates are auto-correlated in time so that each gray dot is not independent of every other dot.

Could more details regarding computation of confidence intervals be provided, e.g. are they symmetric?

To address this, we have updated the Methods as follows:

Methods, after Equation (10): “The SSPE method also tracks the state estimation error (Ptt), which allows estimation of confidence in the mean phase estimates. To do so, we sample 10,000 samples from the posterior distribution of each state j, and from this distribution estimate the width of the 95% credible interval for the phase. The credible intervals are derived from the posterior distribution of the state, which is Gaussian; however, following their transformation into angles, the intervals are not constrained to be symmetric. Note that, alternatively, one could attempt to define a closed form for the phase distribution to estimate confidence bounds …”

Can the credible intervals be related to error in degrees?

Yes, the credible intervals are expressed in units of radians. We have updated Figure 6 to units of degrees.

They look pretty negligible in the left part of the figure (given that the real-time procedure also introduces a lag, Figure 8B),is the plotted x-range the relevant one?

The real-time procedure in C++ introduces a small lag due to buffering implemented in OpenEphys. This lag is not present if the user does a non-buffered analysis as provided in the MATLAB implementation.

How does SNR/amplitude relate to phase accuracy to credible intervals?

To address this question, we now include in the revised manuscript a new simulation comparing SNR, credible interval width, and phase accuracy. Please see R1.A1 for details.

If the spread of credible intervals can be largely explained by amplitude (aka high correlation between credible interval width and amplitude), then maybe this is not such a big factor?

To address this question, we have updated the Results as follows:

Results: Confidence in the Phase Estimate: “… Therefore, a fixed amplitude threshold does not guarantee an accurate phase estimate. Instead, we propose that inspection of the credible intervals better guides certainty in the phase estimate. These credible intervals are directly derived from the modeling procedure, as opposed to surrogate measures of confidence in the phase estimate, such as amplitude. In addition, credible intervals are directly interpretable unlike amplitude thresholds, which depend on the measurement scale (e.g., a credible interval threshold of 10 degrees versus an amplitude threshold of 10 millivolts). We conclude that amplitude is linked to confidence in the phase, however, a fixed amplitude threshold is not directly interpretable and encompasses a wide range of credible interval values.”

2) Slow driftsA practical reason why existent algorithms rely on buffering is the frequent existence of slow drifts in the data, for which detrending needs to be applied. For the EEG data, the following methodological detail of the current approach is given "Before real-time phase estimation, we high-pass filter the data at 0.5 Hz to remove slow drifts." This procedure is not possible when estimating the phase in real-time. The authors claim the benefit of their approach is the absence of needing buffers, but in the light of this I am not sure about whether this is the case. In the discussion, the authors discuss the possibility of modeling the slow drifts via an additional rhythm in the state space, but state that long segments of data are needed for this. In my view, this is a crucial aspect of any algorithm targeting real-time phase and more information and analysis resolving slow drifts would greatly improve the actual applicability of the present approach.

R3.A2. To address this important issue, we have updated the procedure to apply a real-time method that avoids acausal filtering. We have updated the Methods to describe this new approach:

Methods: in vivo Data: “…We analyze 250 s of data from a single participant for demonstration purposes. For these data, before each real-time phase estimation step, we first apply a moving average causal filter (using 3 s of past data for each new data point) to remove slow drifts. We track three oscillators in the SSPE method, …”

Discussion: ”Alternatively, the SSPE model could be updated to include a model of the slow drift dynamics (perhaps as a piecewise linear term), or a moving average mean trend could be causally removed as we demonstrated with the in vivo EEG analysis…”

Using this new real-time approach, we find results consistent with the original manuscript. We include the updated Figure 8 (previously Figure 7) and revised caption here:

3) Data generation process and simulationsI find some of the simulations not sufficiently motivated from a physiological perspective.– Two simultaneous rhythms with nearby frequencies: In this scenario the authors simulate the signal as a summation of two sinusoids with a frequency offset with a fixed phase shift. This will result in a signal with amplitude modulation with the envelope given by the difference of the frequencies, resulting in the appearance of oscillatory bursts. I would argue that for physiological signals that present like this, it is not of interest to track a single frequency in this signal, as e.g. the trough phase of this compound signal corresponds to a specific physiological state. I would also suggest showing the amplitude for this case in Figure 3. Possibly the simulation could be modified to be better physiologically motivated, as they are certainly cases where there are neighboring rhythms which present on the same electrode through volume conduction, but it is unclear (but would be of interest) to me if the proposed method could be helpful in this case.

R3.A3. We appreciate the Reviewer’s concerns regarding this simulation. Indeed, the Reviewer’s interpretation is correct: the summed sinusoids can be interpreted as a single neural population oscillating at the average of the two frequencies, with amplitude modulation. With that interpretation, a different analysis approach – perhaps focused on cross-frequency coupling – might be preferred, However, we note that multiple interpretations exist for such a situation, i.e., we believe the inverse solution here is degenerate making our prior expectation important. In many experimental situations, limited spatial resolution prevents distinguishing whether an observed signal represents a single neural population or rhythms from distinct neural populations. In addition, we expect situations in which multiple rhythms exist at nearby frequencies (such as multiple α in EEG over sensorimotor cortex or multiple theta in the hippocampus). Consistent with the latter examples, in this simulation, we assume the prior expectation that the observed signal results from two rhythms.

We have added text to clarify this assumption, and added the observed signal to the revised Figure as suggested:

Methods: Two Simultaneous Rhythms With Nearby Frequencies: “… In many physiologic use cases (e.g., in EEG (VanRullen, 2016), and in LFP (Tort et al., 2010)), we may wish to track one rhythm while ignoring another at a nearby frequency. This may occur because of low spatial resolution (e.g., from mixing of signals due to volume conduction (Nunez and Srinivasan, 2006)) or because there exist overlapping neural populations expressing different rhythms (Kocsis et al., 1999). To simulate this case, we consider two sinusoids. …"

– From my perspective phase is only well-defined if there are clear-cut oscillatory bursts. In that sense, the signal given as an example in Figure 2A iv does not have clear-cut oscillatory states, with no clear spectral peak and the true phase (blue curve) changing rapidly. The authors show that the algorithm performs well here as this is exactly according to their data-generating process assumptions (e.g. Figure 2C right-most), whereas for the other algorithms errors accumulate especially in fast phase changing periods. In my view, this simulated situation conceptually does not capture processes that are relevant to investigating neural oscillations in the brain, which would follow a conceptual model more similar to: multiplicative 1/f contribution * oscillator + additive 1/f contribution, so that presumably corresponds to the presented filtered pink noise case here. Can the authors clarify the motivation for this simulated case?

R3.A4. In the revised manuscript, we now better motivate the state-space model in terms of neural oscillations in the brain. Please see R2.A3 above.

Additionally, we simulated the model suggested by the Reviewer in two ways. In the first, we implemented a model where (1/f) noise was simulated and directly multiplied by a cosine oscillator at 6 Hz, sample by sample. We plot an example realization of this process in Author response image 1:

**Author response image 1. sa2fig1:** 

Sign changes in the pink noise result in discontinuous jumps in the signal. Given this, we attempted a second simulation where the multiplicative (1/f) component was understood to mean that the amplitude had a power spectrum that followed a (1/f) trend, while the phase followed a 6 Hz rotation, with added (1/f) noise. We plot an example realization of this process in Author response image 2:

Testing the error in phase estimation for this signal with added pink noise, we find that the acausal FIR, Blackwood and SSPE methods perform comparably, while the Zrenner approach performs slightly worse than the others (see Author response image 3).

**Author response image 3. sa2fig3:** The SSPE method performs as well as the acausal FIR and Blackwood methods for 1/f amplitude modulated data.

We include this result here for the Reviewer, and if recommended, we will update the manuscript to include it.

– The authors speak of broadband oscillations, for instance relating to the mu-rhythm (l. 554). In my view this is a narrowband oscillation with a clear peak judging from the power spectrum, with the broadness of the peak influenced by the presence of strong amplitude envelope modulations. Can authors clarify their criteria for defining an oscillation as broadband, in my view this is non-standard terminology.

R3.A5. As suggested by the Reviewer, we now refer to the mu rhythm as narrow-band:

Results: Example in-vivo application: human EEG: “… Like the theta rhythm in the LFP rodent data, the mu rhythm in the human EEG data maintains a consistent spectral profile in time (Figure 8B) and further, it appears to be narrow-band (Figure 8C).”

Figure 8, caption: “(C) Average power spectral density (PSD) of (B) over all times in (A). The mean PSD (dark blue) and 95% confidence intervals (light blue) reveal a peak in the mu rhythm near 11 Hz. (D) Example human EEG data (red, solid) with a consistent peak in the mu rhythm (8-13 Hz). …”

– One limitation of band-pass filtered approaches that the authors mention is non-sinusoidality (l. 48). In my view, this is manifested in differences in waveform shape, e.g. the arc-shape of the mu-rhythm in Figure 7D. It does not seem that problems in phase estimation for this type of signals (e.g. non-uniform phase-velocity) is addressed by the method as the estimated state does not capture the arc-shape of the rhythm. Can the authors clarify in what way they contribute to resolve this limitation?

R3.A6. We thank the Reviewer for this nice observation. For the human EEG, the SSPE method models both the 10 Hz and the 22 Hz (a potential harmonic) rhythms that together capture the non-sinusoidality of the mu-rhythm. In Figure 7D of the original manuscript, we only showed the 10 Hz component of the model. When we include the summation over the (real-valued portions of the) state estimates of both rhythms, the model estimates capture the arc-shape of the mu rhythm (see Figure 8—figure supplement 1). We also show the phase for the two oscillators, demonstrating the capability of the SSPE approach to estimate simultaneously the phase of the 22 Hz harmonic and the 10 Hz rhythm.

We have updated the manuscript to include this result as a Supplementary Figure as follows:

Results: Example in-vivo application: human EEG: “As with the LFP, we are able to assess certainty in our phase estimates using the credible intervals. In Figure Supplement 8-III, we show the state estimate including both the 10 Hz and 22 Hz components of the model and demonstrating that it recreates the non-sinusoidal waveform of the mu rhythm.…”

4) I would welcome additional details in the text regarding following methodological procedures, which should yield a more self-contained manuscript:– While bandpass filtering and Hilbert-transform are quite common operations for electrophysiologists, a state space approach may be unfamiliar to the typical reader. Possibly it would be helpful to have clearer paragraphs describing the framework (aka rework l. 98ff), I think the attempt here was to first explain the procedure without formulas, which appear further down, I am not quite sure that this succeeded in providing clarity. Maybe it would be helpful to first make an example with only 1 rhythm, so that dimensionality of the involved entities is not as complex as for tracking N rhythms. Maybe it would also be helpful to extend Figure 1 with a conceptual diagram. How is the number and frequency of oscillators chosen for empirical data?

R3.A7. As recommended, we have update Figure 1 to include a conceptual diagram to help improve understanding of the SSPE method:

Further, we have edited the beginning of the Methods section to improve accessibility, and now include an example with only 1 rhythm, as recommended:

Methods: State Space Model Framework:

“To estimate phase in real-time, we utilize a data-driven model that operates based on principles from dynamical systems and Markov models. This type of model (called a state-space model) separates what is observed (called the observation equation) from what we wish to estimate (called the state equation). The state equation attempts to capture the underlying, unobserved dynamics of the system, while the observation equation transforms the state into the observed signal. We generate an optimal prediction for the state using a Kalman filter that compares the initial prediction of the observation with the actual observed value. […] Note that this model for the state is akin to a damped harmonic oscillator driven by noise, a model that has been shown to be relevant and useful for γ rhythms in the visual cortex (Burns et al., 2010; Spyropoulos et al., 2020) and also for EEG (Franaszczuk and Blinowska, 1985).

To model N rhythms, we define …”

How is the number and frequency of oscillators chosen for empirical data?

As we discuss in the Methods section, we chose the number and frequency of oscillators for the empirical data based on a preliminary analysis of the power spectral density. We have reproduced the relevant text below:

In vivo data: To model these data, we fit three oscillators with the SSPE method, with the goal of tracking a target theta rhythm (defined as 4 – 11 Hz). Motivated by an initial spectral analysis of these data, we choose one oscillator to capture low frequency δ band activity (ωj=1 Hz, based on spectral peak), one to capture the theta band rhythm of interest (ωj=7 Hz), and a final oscillator to track high frequency activity (ωj=40 Hz).

– E.g. regarding the parameter estimation the reader is referred to the supplementary material of Soulat et al. 2019. Can an intuition about this procedure be given? I find this interesting in the context that a quite large range for expected computation time is given (2-100 s, l. 573). Can the authors comment on factors influencing this? Maybe a more detailed description of the procedure will be helpful for understanding this.

R3.A8. To address these questions, we have added text to the Methods section to help further explain the EM procedure and offer potential reasons for the range of times to convergence:

Methods: Real-time Phase Estimation:

“We perform real-time estimation of phase as follows. First, we use an existing interval of data to acausally fit the parameters of the state space model aj, ωj, Qj, σR2 using an expectation-maximization (EM) algorithm as proposed by (Soulat et al., 2019); for details see the Supplementary Information in (Soulat et al., 2019). Under the EM approach, optimization of parameters follows a two step process. First, initial values for parameters are selected, usually from prior knowledge such as an examination of the power spectrum of an initial data sample. This allows the algorithm to estimate the expectations for the state and the observation. Second, using the state and observation estimates, an analytic solution exists (Shumway and Stoffer, 1982) for the parameters that maximizes the likelihood. We repeat the expectation and maximization procedures until the parameter estimates do not change between two iterations beyond a threshold. The rate of convergence of the EM algorithm (speed of computation) depends on the initial parameter estimates and the signal-to-noise ratios of the rhythms present in the signal.”

– Currently descriptions of the comparison algorithms are lacking, e.g. what is the general procedure and difference between Zrenner and Blackwood algorithm, what are the crucial parameters here? The authors give code-accessibility as the main reason for selecting those two specific instances. Because recently a lot of different methods (mostly variants involving band-pass filtering) were proposed for estimating the phase in real-time, maybe those 2 selected ones could be better discussed in the context of others in the literature.

R3.A9. Thank you for this suggestion. We have updated the manuscript to place the two algorithms better in the context of others in the literature as follows:

Methods: AR-based forecasting to Estimate Phase: “While current techniques depend upon filtering to isolate a narrow band rhythm from which to estimate phase, different methods exist to forecast the phase. One category of methods forecasts the phase as a linear extrapolation from the current moment while assuming a singular central frequency, which is estimated from the data (Rivero and Ditterich, 2021; Rutishauser et al., 2013). Other methods estimate phase by using an autoregressive (AR) model. After estimating the AR parameters (either asynchronously or synchronously), these methods forecast data to limit filter edge effects, then utilize a filter and Hilbert transform to estimate phase (Blackwood et al., 2018 and Zrenner et al., 2020). We compare the SSPE to this latter set of techniques as these are more flexible to variability in rhythms than methods that assume a constant central frequency when estimating phase. We implement the version of the Zrenner et al. (2020) algorithm accessible at …”

We have also added information about the critical hyperparameters of the Zrenner and Blackwood approaches:

Methods: AR-based forecasting to Estimate Phase: “ … We implement the version of the Zrenner et al. (2020) algorithm accessible at https://github.com/bnplab/phastimate and term this method as Zrenner. Please refer to Zrenner et al. (2020) for algorithm details. The Zrenner algorithm uses an FIR filter (applied forward and backward) and the Hilbert transform as the core phase estimation technique. The AR model used to forecast the data and allow real-time phase estimation (by expanding the window for the FIR filter) is fitted to the 30th order and efficiently updated on every new sample. We set the algorithm to track a 6 Hz rhythm by increasing the window size for the FIR filter to 750 ms, the filter order to 192, and the frequency band to 4 to 8 Hz. We utilize the implementation of the Blackwood et al. (2018) algorithm as provided by the authors. In Blackwood et al. (2018), the authors develop a forward bandpass filter – AR-based (5th order AR model whose parameters are fit every second) forecasting and then the Hilbert transform – to compute the phase. Here, we instead apply a Hilbert transformer (of order 18) that estimates the phase by directly computing the Hilbert transform while filtering the data as implemented in (Blackwood, 2019). We do so because this implementation is more computationally efficient, functionally equivalent to the original method and the code is made widely available. We verified in sample data that application of the original and modified Blackwood methods produced consistent phase estimates. We refer to this method as Blackwood.”

– typo, l.55 extant -> existant

R3.A10. Thank you, we have edited the manuscript, replacing “extant” with “existing”.

Introduction L55: “Many existing approaches depend on buffered processing (e.g., Fourier or related transforms), …”

– the dotted lines for phase estimates may be not optimal visually (e.g. Figure 6B), maybe shading of credible intervals would look nicer?

R3.A11. Thank you for the suggestion. We have updated the figures to use shading in the revised manuscript. We show the relevant components of the revised figures here:

– Can the authors provide more details on how the power spectra were estimated? The simulation ones look like Welch's method and the data ones like multitaper were used.

R3.A12. As recommended, we have edited the caption of Figure 2 to include these details:

“… (B) For each scenario, example spectra of (B.i) the signal, and (B.ii) the observation (i.e., signal plus noise). Spectra were estimated for 10 s segments using the function “pmtm” in MATLAB, to compute a multitaper estimate with frequency resolution 1 Hz and 9 tapers.”

For the in vivo data, the Figure captions include details of the power spectrum calculation, reproduced here:

Figure 6 “… (B) Corresponding spectrogram of the rodent LFP data (multi-taper method with window size 10 s, window overlap 9 s, frequency resolution 2 Hz, and 19 tapers). (C) Average power spectral density (PSD) of (B) over all times with theta band oscillator identified in (A). The mean PSD (dark blue) and 95% confidence intervals (light blue) reveal a broadband peak in the theta band.

Figure 8 “… (B) Corresponding spectrogram for the EEG data (multi-taper method with window size 10 s, window overlap 9s, frequency resolution 1 Hz, and 19 tapers). (C) Average power spectral density (PSD) of (B) over all times in (A). The mean PSD (dark blue) and 95% confidence intervals (light blue) reveal a broadband peak in the mu-rhythm near 11 Hz. …”

– If I understood correctly, the employed two-sided t-test is calculated using lots of correlated data points (using all time points in an interval), resulting in reduced degree of freedom. I think the evidence that the SSPE approach works better here is actually strong, but the t-test looks slightly weird because of wrong degrees of freedom, distracting from the result. Maybe the information from table 1 can just be plotted in a distributional way, more like Figure 2C?

R3.A13. The statistical test compares the error estimates across all simulation iterations (N=1000), wherein each iteration has an independent source of noise. Error within each iteration is calculated using all time points within one cycle of the phase reset. We note that the error distributions for each method are approximately normally distributed (see Author response image 4 and 5), and we find that plotting these error estimates as distributions is rather uninformative (Author response image 4) given the tight bounds of the error distributions, and the large differences in means. We therefore think a table is a better approach for conveying this result.

**Author response image 4. sa2fig4:** 

We also examined the error to confirm approximate Gaussianity (Author response image 5):

**Author response image 5. sa2fig5:** 

[Editors' note: further revisions were suggested prior to acceptance, as described below.]

Reviewer #3 (Recommendations for the authors):I thank the authors for extensive replies to my comments, which have clarified many aspects and made the method description more accessible. I like the new waveform shape figure showing the simultaneous tracking of the base + harmonic frequency. I find the SSPE method interesting from a principal perspective, but am not yet sure in what way the performance of the algorithm is superior to other algorithms in physiologically relevant cases. My remaining comments:1) Plausibility of simulations R3.Q4– The state space model simulations (Figure 2A.iv) generating model are not a common data generation model for the rhythms the authors actually investigate (theta and mu). The newly added citations are for the γ-rhythm (and a quite obscure EEG paper). -> I would say this is not a generally accepted data generation model for mu and theta. The fact that the SSPE performs well on data generated from the exact state space model does not seem to entirely relevant for the chosen application, with the simulation lacking stability in cycle periods that is present for mu and theta. Possibly, a note can be added in the Discussion section in the narrow-band vs broad-band signal section, reflecting on broad-band phase would be beneficial.

We agree with the Reviewer that (i) the state space model has not been shown to reproduce theta and mu rhythms, and (ii) it is not surprising that the SSPE method performs well on data generated from the same state space model. We propose that the results shown here provide initial evidence that the state space model can produce theta and mu rhythms, consistent with the in vivo data. We also propose that modeling rhythms as damped harmonic oscillators driven by noise is both simple and physically reasonable. Finally, we note while the parameters used for the state space model instantiation in Figure 2 do not produce phase stability, the model can sustain such phase stability, as demonstrated by the intervals of transient phase stability the Reviewer identifies for the mu and theta rhythms (Figure 7,8).

As suggested, we have updated the Discussion to note that the phase cycles of the state space model and filtered pink noise are more unstable:

“Moreover, when the broadband rhythm has non-negligible power across all frequencies (in which case the phase cycle is more unstable), the standard non-real-time approach (acausal FIR and Hilbert transform) tracks the phase with limited accuracy.”

– The filtered pink noise seems to result in a broader spectral peak (green line in Figure 2B.ii) than is present in the empirical data. As the band-pass filter parameters of the alternative algorithms are possibly adjusted for more narrowband signals, not adjusting the bandwidth before e.g., doing a Hilbert transform may yield an unfair comparison. Can you comment on that?

To address this, we now emphasize in the Discussion the importance of filter bandwidth parameters,

“Narrow-band vs Broadband Signals

While brain rhythms tend to be broadband (Buzsaki, 2004; Roopun et al., 2008), the most common methods in neuroscience to track a rhythm’s phase require appropriate choices of filter parameters (e.g., selection of bandwidth) and generally expect a narrowband oscillation. …”

We also note that the SNR in Figure 2B.ii is high, increasing the apparent bandwidth. Increasing the background pink noise (and making the simulated signal more consistent with the in vivo data), would decrease the apparent bandwidth in Figure 2B.ii (see Figure 5 —figure supplement 1).

Finally, we now make public all the code for the simulations and for plotting the results of the simulations so parameters across approaches can be compared. It is available here: https://github.com/Eden-Kramer-Lab/SSPE-paper

– My suggested approach using multiplicative 1/f corresponds to the second of the simulations provided by the authors in R3.A4. To clarify this, I attach some python lines at the end to show a simulated signal, if there is interest (parameters to be adjusted). The phase estimation errors in the provided plot in the response seem really low, so maybe the oscillatory SNR is too high for this simulated case (also the amplitude modulation is not really prominent?) I find it confusing that the SSPE method does not yield any benefit here. I am not requesting specifically for this type simulation to be included, but I would like to understand generally the conditions under which SSPE yields better results than the alternative algorithms.

Thank you again for recommending this interesting model. The SSPE method does not yield any benefit here because the simulated model of phase is a sinusoid. Thus, methods that assume a narrowband oscillation do better, even when the rhythm waxes and wanes via amplitude modulation; after an accurate estimate of phase when the SNR is high, the true and estimated phases progress linearly, even when the SNR is low (i.e., no rhythm is visible). This is most obvious for the Blackwood and acausal FIR approaches, and less so for the Zrenner method which forecasts more data prior to phase estimation.

We provide here results from the new simulation, where we set the SNR = 0.5 (code also available at https://github.com/Eden-Kramer-Lab/SSPE-paper/blob/main/simulations/sim_multOneOverF_amp.m):

**Author response image 6. sa2fig6:** Example data trace for the proposed model. The amplitude of the 6 Hz sinusoid waxes and wanes in time.

**Author response image 7. sa2fig7:** The Blackwood method outperforms the other causal algorithms for the multiplicative 1/f sinusoid example, similar to “Sines in Pink Noise”.

We conclude – based on the existing simulations in Figure 2, and the new simulation suggested here – that these examples illustrate a more complicated version of the “Sines in Pink Noise” simulation shown in Figure 2; when the true signal is a sinusoid, those estimators that presume a narrow band signal (i.e., the filter based approaches) perform best.

From my understanding, SSPE should yield benefits for phase estimation in intermediate SNR ranges, but at the moment the authors have added more evidence that all algorithms show very similar behavior at high SNR (Figure Supplement I for Figure 7 and 8) and the simulation type provided in Figure 5 has some issues with phase stability as it is filtered pink noise (see comments above) and the modelled SNR range (there is not much change in phase estimation errors which I would have expected to vary with increasing SNR).

We propose that the primary benefits of the SSPE method are due to the state model, more so than the SNR. If the state model is inaccurate, then a phase estimation method will often perform poorly across a broad range of SNR. This is illustrated in Figure 5; estimates from the Blackwood and Zrenner methods do not improve with increasing SNR because the narrowband state model is inaccurate (for this simulated broadband rhythm). For the in vivo data examples (Figure 7,8 Supp I), the phase estimates agree when the rhythm of interest is clearly present. At these times, different state models (sinusoids or noise driven oscillators) produce qualitatively consistent phase estimates.

R3.Q3:I can acknowledge the scenario of the two simultaneous rhythms with nearby frequencies which the authors attribute to two different rhythms instead of seeing as amplitude modulated bursts. For this case, maybe the modeling scenario would be more general, if the rhythms had varying phase lag (random for each trial) instead of a fixed phase lag (pi/4), which seems like the case in which the rhythms are most easily distinguishable.

The Reviewer raises a good point. To address this, we implemented this recommendation and repeated the simulations in Figure 3 such that the phase lag between the two rhythms varied randomly (uniformly from 0 to 2*pi) for each trial. We find qualitatively consistent results, Author response image 8:

**Author response image 8. sa2fig8:** . We find that SSPE still performs well for two oscillations at nearby frequencies, with random initial phases; compare to Figure 3 of the main manuscript.

We now state this results in the revised manuscript as follows,

Following Equation (14): “… where [0.2,2] and [1,11]. We note that replacing the constant phase shift (π/4) with a randomly selected (uniform [−π, π]) phase shift produces qualitatively consistent results. The confounding rhythm frequency assumes a range of integer values from 1 to 11 Hz. …”

2) The relationship between credible intervals and amplitude measures. R3.Q1.– The authors argue for a dissociation between credible intervals and amplitude values, with evidence provided by e.g., Figure 7. Given the nature of the data, with oscillations occurring in bursts and not as a continuous oscillation, I am wary of phase estimates for time points where there is no oscillation (e.g., for the theta data, oscillation probability is heavily dependent on animal speed and there will be periods in the data without an oscillation). My main point here is that credible intervals seem to have a correspondence to amplitudes in a range where an oscillation is clearly present (high amplitude) and phase is not defined when an oscillation is absent. Sure, the acausal FIR will return a value for every time point, but it's not necessarily meaningful in a physiological sense. Taking this as the ground truth is problematic for the credible interval and amplitude measures dissociation. For instance, is there a discernible oscillation for the data in Figure 7 for an amplitude criterion > 65th percentile? Is a wide or narrow credible interval around a value that is not defined meaningful?

To address this, we provide here additional examples of the LFP data shown in Figure 7. These examples illustrate cases in which the amplitude is low (below the 65th percentile), but the credible interval is narrow. At these times, visual inspection suggests a theta rhythm exists, despite the low amplitude. These examples illustrate how an amplitude threshold may fail to capture transient bursts of theta. Conversely, when a theta burst exceeds an amplitude threshold, existing methods may be overly confident in the stability of the phase cycle; these examples show that, even when the amplitude is high, the credible intervals can be large.

**Author response image 9. sa2fig9:** Credible intervals better capture the presence of theta cycles than an amplitude threshold. (A,B) Two example traces showing intervals in which the rhythm amplitude is small (below the 65th percentile, white intervals, also indicated by purple trace equaling 0), yet the credible intervals suggest confidence in the phase estimate, consistent with the presence of a theta rhythm. Visual inspection of data filtered (4 – 11 Hz FIR filter, blue curve) in the theta band suggests the presence of a theta rhythm. Gray intervals indicate periods of high amplitude (above the 65th percentile). In these intervals there exist periods with wide credible intervals (at times when the SSPE method is uncertain if the rhythm is rising or falling) suggesting a simple amplitude threshold is insufficient to determine confidence in the phase estimates.

Figure 7C is a scatter plot with lots of points that are correlated, maybe it would be good to visualize this in a different way reducing the number of points per cyclepossibly by subsampling or a 2D histogram. In the current presentation, the plot basically just shows the range of the data, amplifying low-occurrence data points.

As recommended by the Reviewer, we have updated Figure 7C to include a 2D histogram replacing the scatterplot:

This suggests that, at low SNR, an arbitrary amplitude threshold (e.g., amplitude > 1) would eliminate all data at SNR=1, and preserve all data at SNR=2.5.Typically (e.g., in the Zrenner studies), amplitude is defined as a relative criterion, e.g., with percentile threshold or mean amplitude +- 2.5 SD using some training interval, so I am not sure about this argument.

The Reviewer identifies a nice solution to avoid the choice of an arbitrary amplitude threshold eliminating or including all data. However, we propose that this criterion still lacks the direct interpretability of a credible interval threshold (e.g., 30 degrees, which is easily interpreted and may include data across amplitudes).

– We find that fixed amplitude thresholds produce wide ranges of credible intervals (Figure 7C.ii). This is especially apparent at lower amplitude thresholds; choosing a threshold of 65% results in credible intervals that range from 0.042.29 (minimum) to 174.75 (maximum) degrees.Can authors list the 95% range here, rather than min/max?

Thank you for the suggestion, we have updated the test to reflect the 95% range:

“This is especially apparent at lower amplitude thresholds; choosing a threshold of 65% results in credible intervals that range from 3.8 (2.5th percentile) to 96.8 (97.5th percentile) degrees.”

– Figure 5: Can a spectrogram can also be plotted in this figure?I would expect the height of the spectral peak over the 1/f-contribution to vary here.

To address this, we now include a supplement to Figure 5 that includes the spectra for each simulated SNR.

We have updated the manuscript as follows:

“… In these simulations the SNR varies from 1 (signal and noise have the same total power) to 10 (signal has 100 times the total power of the noise); see example traces of the observation under each SNR in Figure 5A.i – iv, and see example spectra under each SNR in Figure 5 —figure supplement 1. We repeat each simulation 1000 times with different noise instantiations for each signal to noise ratio.”

3) Code availabilityI wanted to check the used bandpass filter parameters (since it's not described for the Blackwood algorithm) and noticed that the linked repository does not feature the code to recreate the plots in the manuscript, possibly it would be helpful for other researchers to provide that, also for comparing to other algorithms in the future.

Thank you for raising this important point. We have created an independent repository with code to run the simulations and to recreate the plots in the manuscript. Please see, https://github.com/Eden-Kramer-Lab/SSPE-paper.